# Phytochemicals in Prostate Cancer: From Bioactive Molecules to Upcoming Therapeutic Agents

**DOI:** 10.3390/nu11071483

**Published:** 2019-06-29

**Authors:** Bahare Salehi, Patrick Valere Tsouh Fokou, Lauve Rachel Tchokouaha Yamthe, Brice Tchatat Tali, Charles Oluwaseun Adetunji, Amirhossein Rahavian, Fhatuwani Nixwell Mudau, Miquel Martorell, William N. Setzer, Célia F. Rodrigues, Natália Martins, William C. Cho, Javad Sharifi-Rad

**Affiliations:** 1Student Research Committee, School of Medicine, Bam University of Medical Sciences, Bam 44340847, Iran; 2Antimicrobial and Biocontrol Agents Unit, Department of Biochemistry, Faculty of Science, University of Yaounde I, Ngoa Ekelle, Annex Fac. Sci, Yaounde 812, Cameroon; 3Institute for Medical Research and Medicinal Plants Studies, Yaoundé 13033, Cameroon; 4Antimicrobial Agents Unit, Laboratory for Phytobiochemistry and Medicinal Plants Studies, Department of Biochemistry, Faculty of Science, University of Yaoundé I, Messa-Yaoundé 812, Cameroon; 5Applied Microbiology, Biotechnology and Nanotechnology Laboratory, Department of Microbiology, Edo University, Iyamho, Edo State 300271, Nigeria; 6Department of Urology, Shohada-e-Tajrish Hospital, Shahid Beheshti University of Medical Sciences, Tehran 1989934148, Iran; 7Department of Agriculture and Animal Health, University of South Africa, Private Bag X6, Florida 1710, South Africa; 8Department of Nutrition and Dietetics, Faculty of Pharmacy, University of Concepcion, Concepcion 4070386, Chile; 9Department of Chemistry, University of Alabama in Huntsville, Huntsville, AL 35899, USA; 10LEPABE–Department of Chemical Engineering, Faculty of Engineering, University of Porto, Rua Dr. Roberto Frias, s/n, 4200-465 Porto, Portugal; 11Faculty of Medicine, University of Porto, Alameda Prof. Hernâni Monteiro, 4200-319 Porto, Portugal; 12Institute for Research and Innovation in Health (i3S), University of Porto, 4200-135 Porto, Portugal; 13Department of Clinical Oncology, Queen Elizabeth Hospital, Hong Kong, China; 14Zabol Medicinal Plants Research Center, Zabol University of Medical Sciences, Zabol 61615-585, Iran

**Keywords:** prostate cancer, medicinal plants, phytotherapy, secondary metabolites, plant formulas

## Abstract

Prostate cancer is a heterogeneous disease, the second deadliest malignancy in men and the most commonly diagnosed cancer among men. Traditional plants have been applied to handle various diseases and to develop new drugs. Medicinal plants are potential sources of natural bioactive compounds that include alkaloids, phenolic compounds, terpenes, and steroids. Many of these naturally-occurring bioactive constituents possess promising chemopreventive properties. In this sense, the aim of the present review is to provide a detailed overview of the role of plant-derived phytochemicals in prostate cancers, including the contribution of plant extracts and its corresponding isolated compounds.

## 1. Introduction

### 1.1. A Brief Overview on Prostate Cancer

The rapid growth of chronic diseases over the past century, including cancers, has emerged as among the most difficult situations for public health systems in underdeveloped and developing countries [1]. Cancer is one of the most prominent health issues in all countries due to its growing prevalence, mortality rate and high treatment cost in both genders and in all ages. In general, cancer remains not only a cause of tremendous damage to health but also the second leading cause of morbidity worldwide [2,3].

Cancer is caused by uncontrolled cell proliferation that can take place in different tissues and spread into surrounding and distant tissues [4]. Despite the main progress made in cancer biology, cancer remains one of the principal causes of mortality, and those who survive can experience permanent complications (e.g., physical, cognitive, psychosocial struggles, and treatment side effects) [5,6]. It is of great concern to note that cancer is a widespread disease and diagnoses are sharply increasing globally. Many risk factors are mentioned for this rise and lifestyle changing play the most important role [7].

Oncology studies have shown several types of cancer that are commonly diagnosed, including prostate, lung/bronchus, colorectal, breast, stomach, and liver cancer [8]. Although there is some variation in cancer prevalence, prostate cancer was the most commonly diagnosed cancer in the United States of America (USA), Europe and Oceania in 2012 [9]. In the past decade, much attention has been focused on prostate cancer [10] due to the alarming number of patients and the high mortality rate [11,12]. In fact, prostate cancer is the second most deadly malignancy in men after skin cancer [13]. Also, it is the most frequently diagnosed cancer among men, with a high mortality rate. About 1.6 million new cases of prostate cancer were diagnosed in 2015, and 366,000 deaths were reported [14]. In comparison to 2012, there was an increase of about 45% in incidence and 19% in mortality rate [15,16,17,18]. According to the American Cancer Society, the risk of cancer diagnosis in men in their lifetime is 1 in 9, and about 1 man in 41 will die due to prostate cancer [19].

The prostate is a glandular organ found under the bladder composed of epithelial cells arranged in a fibromuscular stromal network [20]. Although it has been difficult to establish the definitive etiological clues linking prostate cancer development to incidence, several studies have consistently linked the disease with common risk factors, namely age, race, dietary and physical activity [8,21]. Prostate cancer incidence is, in essence, influenced by age since the risks of being diagnosed with it increases with age [22]. Apart from age and race, Attard et al. [10] have reported that family history, for example a first-degree relative (e.g., father, son, or brother) with prostate cancer has surfaced as the greatest risk factor. According to Pandey, et al. [23], either genetic or somatic mutations contribute 10% or less to the causes of prostate cancer, whilst the remaining 90% has been attributed to epigenetic changes such as lifestyle. However, it is evident that a process that associates risk factors with cancer is inflammation [24]. In order to understand the significant role of inflammation in cancer, it is important to unpack the physiological and pathological processes attributed to inflammation.

Early detection of prostate cancer, like other malignancies, is important for better management and to prevent mortality and reduce morbidity rates, so many studies have been conducted to evaluate the risk of prostate cancer based on signs and symptoms [25]. Some of them have concentrated on lower urinary tract symptoms (LUTS) like hesitancy, nycturia, urinary retention and frequency, but almost all of them concluded that there are no signs and symptoms that are highly predictive of prostate cancer [26] and because it is vital for primary care providers and family physicians to suspect prostate cancer in patients who developed LUTS, it recommended that prostate-specific antigen (PSA) screening, but also digital rectal examination (DRE) should be performed for all of these patients and if any abnormalities detected, patients should be referred to urologists for complementary work-up and distinguishing between prostate cancer and benign prostatic hyperplasia [27,28].

PSA measurement was introduced in 1987 to verify the response to prostate cancer treatment, but was soon adopted for prostate cancer screening too [29,30] and after widespread use of PSA as a screening test, a dramatic rise in incidence was reported from 1989 to 1992 and from 1995 this rise continued with a slight slope until 2001 and after that has fluctuated year to year revealing changes in screening practices [31]. After prostate cancer screening with PSA started in 1991 mortalities have declined and this may be due to early detection and proper management of patients [32]. The cut-off point of 4.0 ng/mL was considered for PSA screening and studies have shown that with this threshold the negative predictive value of PSA for detecting prostate cancer is 89% in men with a median age of 69 years [33], so patients with PSA levels >4 ng/mL in two tests should undergo other work-up like prostate biopsy, multiparametric MRI [34] and whole body bone scans [35]. On the other hand, PSA is not entirely specific for prostate cancer, and other conditions, such as prostatitis, urinary tract infection (UTI), older age, benign prostate hyperplasia (BPH) and bicycle riding can cause elevations in PSA levels, and some medications, like 5α-reductase inhibitors, aspirin, thiazide and statins cause decreases in PSA levels [13,36]. Furthermore, most prostate cancers are not harmful if not diagnosed and treated, and using PSA for diagnosis for prostate cancer results in over-diagnosis and over- treatment, so nowadays there is a vigorous debate about the usefulness of PSA screening for early detection of prostate cancer [37,38]. As a result, researchers have introduced other biomarkers for prostate cancer, such as free PSA, human kallikerin 2, prostate cancer antigen 3, prostate-specific membrane antigen, etc. [39] to better diagnose prostate cancer and avoid over-diagnosis and over- treatment, but there is a public consensus that with evaluation of patient risk factors, physicians can separate high-risk patients and focus on them to not miss any significant cancer, in addition, to decline over-treatment and diagnosis [14,40].

Prostate cancer is a heterogeneous disease, so to anticipate the behavior of cancer, evaluating risk factors is very important [41]. Epidemiological studies have consistently emphasized the notion that naturally-occurring dietary agents possess chemopreventive properties and could easily suppress several malignancies, including that of the prostate [15]. However, there has been an inconsistency regarding a recommended plant-based diet, related nutrients, phytochemicals and prostate health [42]. In this sense, the aim of the present review is to provide a detailed overview of the physiopathology of prostate cancer, including the main risk factors and current therapeutic strategies, and of the role of plant-derived phytochemicals, including plant extracts and its corresponding isolated compounds, in prostate cancer.

### 1.2. Prostate Cancer: Main Risk Factors

The main risk factors can be stratified into two groups: non-modified and modified factors. Non-modified factors are age, family history, ethnicity, and genetic factors [40].

#### 1.2.1. Non-Modified Risk Factors

*Age*: Before 40 years of age, mens’ risk of developing prostate cancer is low. On the other hand, men older than 55 years of age have 17 times more risk of developing prostate cancer than men <55 years old [43]. The mean age when prostate cancer is detected in the United States is 66 years old [44].

*Ethnicity*: Incidence (60%) and mortality rate (2.4 times) of prostate cancer in African-American men is higher than for other races, and Hispanic men, Asian/Pacific Islanders, American Indian/Alaskan Natives are in lower risk of developing prostate cancer [44] and it has been shown that prostate cancer incidence in men who immigrate to regions with higher prevalence rate, is higher than men in their country of origin and this increase depends on the length of stay in that area [45,46].

*Family and genetic factors*: Patients with a positive prostate cancer family history have a higher risk of having this disorder than others, especially a positive history among first degree relatives and in multiple relatives and under 65 years old [47]. Until now more than 105 loci show that increased risk of prostate cancer have been identified, suggesting about 30% of heritability [48,49]. Table 1 showed the relative risk of a family history of prostate cancer [47,50].

*Height*: Another factor that increases prostate cancer risk is height. Taller men are in greater risk of progressive prostate cancer, not total prostate cancer [51] and an overall relative risk of 1.19 has been estimated for prostate cancer per 10 cm increase in height [52].

#### 1.2.2. Modified Risk Factors

*Obesity*: There is no clear relationship between obesity (body mass index (BMI) >25 kg/m^2^ [53]) and increased risk of prostate cancer, but it is proven that obese men are at higher risk of advanced prostate cancer and biochemical recurrence [54,55], and also recent studies showed that risk of recurrence in patients who have weight gain after radical prostatectomy (RP) is higher [56]. Risk of advanced prostate cancer is six times higher than for non-obese men [43] and the risk of mortality increases by 20% for every 5 kg/m^2^ increase in BMI [54]. The importance of this issue is highlighted by the fact that we know that the world’s obese population has at least doubled since 1980 [57] and this can be due to lifestyle changes of patients that have resulted in lower physical activity and higher fat and red meat intake. Physical activity, especially vigorous activity, decreases prostate cancer risk, advance prostate cancer, mortality and recurrence of prostate cancer, and increases survival and it has been shown that physical activity for at least 3 h/week, even jugging and brisk walking, decreases cancer-specific mortality rate [58,59,60]. On the other hand, an inactive lifestyle has been related to higher PSA [60]. Many studies have revealed that higher intakes of fat, red meat, and dairy foods increase the risk of prostate cancer, but it is not proven yet. Dairy products contain a lot of fat and calcium, and high consumption of calcium increases the risk of prostate cancer, and this is probably due to disturbance of the metabolism of vitamin D [61], but non-dairy calcium intake does not change prostate cancer risk [62]. A 2012 study showed that high amounts of red meat and dairy foods elevate the prostate cancer risk 12-fold [63], and there is no specific amount for daily calcium intake, but some studies revealed that consumption of calcium >2000 mg/day raises the risk of prostate cancer [64].

*Infectious disease*: Infections and chronic inflammation leading to cellular damage and hyperproliferation cause 16% of worldwide malignancies [65] and some studies have revealed that UTI, sexually transmitted diseases and prostatitis could cause the development of prostate cancer via this mechanism, but it is uncertain [40,66]. At present, no specific infectious agent has been proven to cause prostate cancer. However, some evidence for the role of *Trichomonas vaginalis* in prostate cancer has been shown [67].

*Occupational and external exposure*: some jobs have a higher risk of prostate cancer due to exposure to specific materials, for example farmers who are exposed to pesticides and other chemical materials have a two times higher risk of prostate cancer [18,68] and also higher exposure to sunlight due to UV and ionizing radiation is related to an increased risk of prostate cancer [69,70].

*Smoking*: Cigarette smokers have a higher probability of developing prostate cancer, including advanced and hormone resistant forms, spreading metastasis and higher mortality rates and it depends on the amount (pack/year) and duration of smoking and it showed that the risk of mortality and recurrence of prostate cancer in former smoker patients, who quit smoking 10 years before diagnosing prostate cancer is similar to that of non-smoking patients [58]. Some researchers are interested in the association among prostate cancer and alcohol intake, and many studies on this topic have been done, but mixed results were obtained, although one case-control study revealed that heavy drinkers have lower PSA levels and are in higher risk of advanced disease at detection [71,72].

*Endogenous hormones*: Androgens cause the proliferation and differentiation of the luminal epithelium of the prostate and play a key role in prostate carcinogenesis and establishing cancer, and because of these facts many patients respond to androgen deprivation treatment. For a long time, researchers believed that high serum androgen level was a risk factor of prostate cancer, but the last pooled analysis could not find any link between prostate cancer and serum androgen levels, but it found a connection among sex-hormone-binding globulin serum concentration and cancer risk [73]. Previously estrogens were a choice of treatment in castration-resistant prostate cancer and have been considered as a protective agent for cancer, but recently more studies have presented evidence for a pro-carcinogenic effect of estrogen on prostate cancer and shown that early exposure to estrogens increases the risk of later prostate cancer [74,75]. A pooled analysis in 2008 showed a strong connection between insulin-like growth factor-I and the risk of prostate cancer [76], but epidemiologic studies reviewed in 2011 revealed mixed findings, although they suggested that the insulin-like growth factor axis affects cancer progression rather than initiation [77]. The core genetic changes that cause activation of oncogenes and inactivation of tumor suppressors are responsible for the start and progression of prostate cancer, and epigenetic and structural genomic changes like deletion, chromosomal rearrangement, and amplification that result in gene fusion with new biologic functions are responsible for these changes. Chromatin remodeling, hypomethylation and promotor methylation that cause epigenetic regulation of gene expression play a significant part in the development and evolution of prostate cancer. Androgen receptors (AR) play a key role in prostate cancer, and changes in ARs like amplification, mutations, and ligand promiscuity are determining factors in progressive castrate-resistant prostate malignancies because these changes sensitize the ARs to low levels of intra-tumoral androgen [78]. The basic drivers for the initiation of prostate cancer are based on gene fusions of TMPRSS2 and the ETS family oncogenic transcription factors [79].

## 2. Therapeutic Strategies: A Brief Summary

To properly treat prostate cancer, patients should undergo full evaluation, including DRE, checking PSA and LFT, life expectancy and comorbidity evaluation, abdominal-pelvic CT, MRI and radionuclide bone scans if needed, and based on these data and characterizations of tumor (Table 2), including clinical stage, Gleason score, tumor volume, invasion and metastasis, patients are stratified into low, intermediate, high and very high risk groups and the cancer divided to localized, locally advance and metastatic prostate cancer [80,81,82].

There are some established options for treating prostate cancer, like watchful waiting (WW), active surveillance (AS), radiation therapy (RT), hormone therapy (HT), and radical prostatectomy (RP) [80]. The goal of conservative management (AS, WW) is to reduce over-treatment [81]. In WW, patients are followed until new symptoms appear or get worse [80], so WW is suitable for poor prognosis patients with low life expectancy [81]. AS is suitable for low-risk prostate cancer or patients with <5 years life expectancy and in AS, physicians monitor patients closely and some periodic work- ups like DRE, PSA checking, prostate biopsy, and MRI are done, and every time the evidence is in favor of cancer progression, patients then become candidates for other definite treatments [80,84]. RP is the first option introduced for treating prostate cancer [85], and it remains a typical form of management because it is the only method that cures the prostate cancer and the goal of RP is to eradicate cancer while conserving urinary continence and if possible potency [81]. Patients with intermediate and high-risk prostate cancer and life expectancy > 5 years are good candidates for RP, and RT is an option for managing almost all prostate cancer groups alone or with another modality, except got low and intermediate risk prostate cancer patients with low life expectancy (<5 years) [84]. RT and RP are the most common methods for managing prostate cancer, and so far, no study has establish the superiority of one of these two methods over the other and complications in both methods are common, and also there are no significant differences between the survival rates of these two methods [80]. There are different approaches for RP, including perineal, retropubic, laparoscopic and robotic, but until now there is no clear evidence that any one of this methods is better than the others in cancer control, cancer-related urinary continence and erectile function conservation, although some poorly designed studies have revealed that robot-assisted RP is better than laparoscopic methods in reducing positive surgical margins [86]. The most popular methods for RT that could be accompanied with HT are external beam radiotherapy and brachytherapy that have side effects like rectal and bladder toxicity and these side effects are more common in external beam radiotherapy. Other treatments like cryoablation and high-intensity focused ultrasound ablation have been introduced, but there is no proof to support their superiority [83,87]. Finally, physicians should choose the proper treatment based on tumor characterization and the patient’s condition after the acceptance of the patient [88].

Many prostate cancer patients have more progressive disease, and management of these patients is different. In patients with symptomatic non-metastatic prostate cancer who are not candidates for curative treatment and patients with symptomatic metastatic prostate cancer, androgen deprivation therapy (ADT) is an option for palliative therapy, but we should not use ADT on patients with asymptomatic locally advanced prostate cancer or biochemical recurrence after curative therapy [82,89]. There are several methods for ADT. The gold standard is bilateral orchiectomy that diminishes the testosterone level below 15 ng/dL on average [90] but this has some disadvantages like irreversibility, physical and psychological pressure on the patients, so HT was introduced [91]. Luteinizing hormone-releasing hormone (LH-RH) agonists (leuprolide, goserelin, triptorelin) and antagonists (degarelix, abiraterone), non-steroidal antiandrogens (bicalutamide, flutamide, nilutamide) are three major drug categories used for ADT with LH-RH agonists being more prevalent, but the risk of flare phenomena is lower when using a LH-RH antagonist [91,92]. Intermittent or continuous ADT are two separate methods for managing systemic prostate cancer, but there is no difference between overall survival and cancer-specific survival of these two methods [82].

It is likely that after any curative management patients eventually relapse, that includes rising PSA or nodal involvement. If patients develop rising PSA after RP, the European Association of Urology guidelines advise early salvage radiotherapy (SRT) [82] and some retrospective studies have revealed that adding ADT to early SRT had some benefits in biochemical progression-free survival after 5 years [93]. In patients with PSA relapsing after RT, salvage RP is the first choice for local control of cancer. Salvage RP increases the risk of anastomotic stricture, urinary incontinence, erectile dysfunction, and rectal injury, so other alternative methods are available, like salvage cryoablation, high and low dose rates brachytherapy [82,94]. For management of nodal relapse, surgical and salvage lymph node dissection (LND) is the only choice. There are no specific criteria for candidate patients for salvage LND, but this should be considered and this method should be used for highly selected patients [82,95].

As we said, patients with the progressive disease can be managed with ADT, but some of these patients develop castration resistance, that is, castrated serum testosterone is less than 50 ng/dL, and the patient has biochemical or radiologic progression [96]. First-line treatment for this situation is abiraterone, enzalutamide or docetaxel (DX)-based chemotherapy and second-line treatment options depend on the chosen first-line treatment. If the patient was treated with abiraterone or enzalutamide as first-line treatment, DX-based chemotherapy is the next option and vice versa. If DX-based chemotherapy was used first and the patient responded, we can repeat this chemotherapy regimen again, but there is usually no improvement in the survival of patients [82,97]. Most of these patients developed with painful bone metastasis, but external-beam radiotherapy is very effective in relieving pain [98]. Finally, it is important to say that managing these patients needs teamwork, and the urologist, oncologist, psychologist, nurse, and even social workers should work together to manage patients properly [99].

Prostate cancer, like the other cancers, is an expensive disease and imposes a great burden on both the health system and patients, and these expenditures are increasing year by year which may due to over-treatment, over work-up or over-diagnosis and increased survival [100]. In 2010, the budget expended for prostate cancer care in the United States was 11.8 billion dollars, and in 2013 and 2017 this budget was $13.0 and $14.8 billion, respectively [101]. In Iran, direct medical costs for prostate cancer were estimated at about 12.5 million USD in 2016 for about 500 patients [102] and the cost for metastatic castration-resistant prostate cancer in Italy in 2016 ranged from €196.5–228.0 million [103]. These cost variations may be due to differences in incidence and management protocols between countries [100], and most of these monies were expended for treatment [103], so having preventive strategies and using natural products for managing prostate cancer patients it is possible to markedly decrease the economic burden of this disease.

## 3. Plant Extracts and Plant-Derived Bioactives in Prostate Cancer

Traditional plants have been used to treat and cure various diseases [104], and this has led to increased use of medicinal plants in the search for new drugs from nature [105]. The discovery of new drugs is often established based on the knowledge that plant extracts can be used to treat diseases in humans. The plants are potential sources of natural bioactive compounds that are, but not limited to, secondary metabolites [106]. Cragg and Newman [107] have stated that any part of a plant such as leaves, bark, flowers, and seeds may contain these secondary metabolites. Although little is known of the primary processes of the secondary metabolites in plants, Bodeker [108] reported that secondary metabolites are essential and important in plant use by people. In this regard, herbal medicines, which have been increasingly used in cancer treatment, represent a rich pool of new and bioactive chemical entities for the development of chemotherapeutic agents with many exhibiting favorable side effect and toxicity profiles compared to conventional chemotherapeutic agents [6,109]. In this sense, in the following section the plant extracts and corresponding bioactive constituents with anti-prostate cancer potential are carefully described. Lastly, a special emphasis on clinical studies confirming the plant-derived phytochemicals anti-prostate cancer potential is also given.

### 3.1. Plant Extracts with Anti-Prostate Cancer Potential

Among the plant extracts with anti-prostate cancer potential (Table 3), the most remarkable ones belong to the Annonaceae, Apocynaceae, Asteraceae, Combretaceae, Euphorbiaceae, Fabaceae, Lamiaceae, Malvaceae, Phyllantheraceae, Poaceae, Rutaceae, Solanaceae and Zingiberaceae families (Figure 1).

#### 3.1.1. Annonaceae Plants

*Annona muricata* or graviola pulp extract (1–5 μg/mL) containing acetogenins strongly inhibited the hypoxia-induced NADPH oxidase (NOX) activity in prostate cancer cells (LNCaP, 22Rv1, and PC-3) viz. 22Rv1 cells by 39–98%, LNCaP cells by 77–91% and in PC-3 cells by 71–75%. This activity was linked to a reduction in the expression of NOX catalytic and regulatory sub-units (NOX1, NOX2, and p47(phox)), nuclear hypoxia-inducible factor (HIF)-1α levels, the proliferative and clonogenic potential. Indeed, NOX expression is directly linked to prostate cancer development in TRAMP mice, suggesting NOX as a possible chemoprevention target in monitoring the disease [118]. Essential oil from leaves of *Anaxagorea brevipes* containing β-eudesmol (13.16%), α-eudesmol (13.05%), γ-eudesmol (7.54%), guaiol (5.12%), caryophyllene oxide (4.18%) and β-bisabolene (4.10%) exhibited high antiproliferative activity on prostate cancer cells, PC-3 (IC_50_ = 9.6 μg/mL) [115]. *Malmea depressa* plants revealed cytotoxic activity in LNCaP greater than 50% [123]. A study exhibited that the methanol extract of *Xylopia aethiopica* has antiproliferative activity in PC-3 and LNCaP cells (IC_50_ = 62.1 and 73.6 μg/mL, respectively) at 96 h through apoptosis, lacking anti-angiogenic properties [172].

#### 3.1.2. Apocynaceae Plants

Luobuma (*Apocynum venetum*), a popular beverage in Asia, has bioactive compounds including sterols (lupeol, stigamasterol, and β-sitosterol) and polyphenolics (isorhamnetin, kaempferol, and quercetin), which were reported for their chemopreventive activity (e.g., anti-androgen-insensitive-prostate-cancer (anti-AIPC)). Fraction 8 inhibited the proliferation of PC-3 cells by arresting G2/M, regulation of apoptotic signal molecules (cytochrome c, Bcl-2, p53, and caspase-3 and -8), inhibition of β-catenin signaling, and suppression DNA repair enzyme expression (uracil-DNA glycosylase) [119]. *Cascabela peruviana* showed an antiproliferative effect on the human prostate cancer cell line (IC_50_ = 1.91 μg/mL). The extract containing thevetiaflavone and cardiac glycosides induced apoptotic cell death, and significantly reduced cell motility and colony formation on all evaluated cancer cell lines [126].

#### 3.1.3. Asteraceae Plants

*Achillea teretifolia* methanol extracts exhibited a remarkable cytotoxic effect on DU145 and PC-3 cells as time and concentration increased, with up-regulation of the mRNA expression level of the pro-apoptotic (bax, caspase-3) and anti-apoptotic (bcl-2), and down-regulation of the expression of bcl-2, and might contain anticancer compounds, activating the cytotoxicity and the apoptosis on prostate cancer cells [112]. *Achillea santolinoides* ethanol extract showed anticancer potential against PC-3 and Du145 cells, with IC_50_ values of 193 and 151 µg/mL, while that of *Hertia angustifolia* displayed an IC_50_ of 216 µg/mL on DU145, respectively [111]. Another study revealed that *Sigesbeckia orientalis* ethanol extract considerably inhibited the proliferation of LNCaP cell lines in 24 h with an IC_50_ = 87.2 ± 1.3 μg/mL [165]. *Wedelia chinensis* has, at least three active compounds—luteolin, wedelolactone, and apigenin—that act synergistically to inhibit prostate cancer cell growth *in vitro*. Tsai et al. [170] reported a standardized optimized herbal ethanolic extract of dried *W. chinensis* (WCE). The combination of LC/MS/MS and PSA reporter assay was suitable to measure the quality and efficacy of a standardized WCE on a xenograft tumor model. Besides, the pharmacokinetics and oral bioavailability of active compounds demonstrate that holistic WCE had extra pharmacological synergy afar the multi-targeted therapeutic effects [170]. WCE was shown to be effective in suppressing crosstalk between the AR and HER2/3 signaling in an in vivo adapted castration-resistant prostate cancer LNCaP cell model that was insensitive to androgen withdrawal and second-line antiandrogen, enzalutamide and offer evidence that the use of a clear, single plant-derived extract can increase the therapeutic efficacy of castration with expressively prolonged progression-free survival warranting further clinical studies [169].

#### 3.1.4. Combretaceae Plants

*Anogeissus latifolia* and *Terminalia bellerica* inhibited the cell proliferation of PC-3 cell lines in a concentration-dependent manner (IC_50_ = 10.6 and 17.7 μg/mL, respectively) [110]. *Quisqualis indica* (QI) reduced (TP)-induced increase in AR and PSA expression in LNCaP. Oral administration of 150 mg/kg of QI together with the TP injection in rat protected against TP-induced BPH, as shown by the decrease decreased prostatic levels of DHT and the anti-proliferative and proapoptotic activities of QI [162]. *Terminalia catappa* also showed anti-proliferative activity in LNCaP with inhibition percentage superior to 50% [123].

#### 3.1.5. Euphorbiaceae Plants

*Baliospermum montanum* is an anticancer plant used in Ayurvedic medicine [167]. Nanoparticles prepared with both aqueous and ethanolic extracts of *B. montanum* presented a dose and time-dependent toxicity on prostate cancer cells with cell viability of 22% and 6% with a maximum concentration of aqueous and ethanolic nanoparticles (2 mg/mL), respectively, in 48 h. Also, no in vitro hemolysis and significant reduction of the wound healing capacity and colony forming ability of the prostate cancer cells was demonstrated [121]. *Cnidoscolus chayamansa* plants showed cytotoxic activity on LNCaP with a percent inhibition greater than 50% [123]. *Euphorbia microsciadia* ethanol extract showed anti-prostate activity against PC-3 cells with an IC_50_ value of 222 µg/mL, and that of *Euphorbia szovitsii* had anti-proliferative activity on both PC-3 and DU145 cells with IC_50_ values of 111 and 56 µg/mL [111].

#### 3.1.6. Fabaceae Plants

*Arachis hypogaea* or peanut skin procyanidins (PSP) and six fractions (PSP-1~6) considerably repressed the proliferation of DU145 cells. PSP-2 consisting in procyanidin B_3_ mainly and procyanidin dimer [(E)C-luteolin or keampferol] secondarily was the most effective fraction that induced apoptosis, cell cycle arrest at S phase, increased intracellular ROS level and decreased Bcl-2/Bax ratio and triggered the activation of p53 and caspases-3 in DU145 cells [120]. *Acacia catechu* inhibited the cell proliferation of PC-3 cell line in a concentration-dependent manner (IC_50_ = 14.3 μg/mL) [110]. *Calliandra portoricensis* (CP) inhibited PC-3 and LNCaP growth between 7% and 92% at 48 h (10, 50 and 100 μg/mL). Detection of cell death induced by CP at 50 μg/mL displayed superior enrichment factors in LNCaP (7.38) than PC-3 (3.48). Moreover, CP (50 μg/mL) notably decreased the network of vessels in CAM, proposing an antiangiogenic potential [124]. Rayaprolu et al. [140] showed in vitro bioactivity of peptide fractions of *Glycine max,* soybeans against PC-3 cells proliferation with up to 63.0% of inhibition (IC_50_ = 608–678 µg/mL). *Glycyrrhiza uralensis* (licorice) extracts are known for their anti-carcinogenic properties but the presence of glycyrrhizin, a hypokalemia and hypertension agent, is detrimental. Park et al. [141] prepared a glycyrrhizin-free hexane/ethanol extract of *G. uralensis* that induces apoptosis and G1 cell cycle arrest and inhibits migration of DU145 cells. *Leucaena leucocephala* plant showed cytotoxic effects on LNCaP with a percent inhibition greater than 50% [123]. *Medicago sativa* and *Sophora alopecuroides* ethanol extracts inhibit the growth of Du145 cells (IC_50_ = 77 and 192 µg/mL, respectively) [111]. *Sutherlandia frutescens* methanol extract (SFE) showed an antiproliferative effect on the human prostate cancer cell lines PC-3 and LNCaP, and mouse prostate cell line TRAMP-C2 (IC_50_ = 167 µg/mL for PC-3, 200 µg/mL for LNCaP, and 100 µg/mL for TRAMP-C2) associated with a dose-dependent inhibition of the Gli-reporter activity in Shh Light II and TRAMP-C2QGli cells treated with SFE. Furthermore, SFE has shown to inhibit Gli/Hh signaling by blocking Gli1 and Ptched1 gene expression in the presence of a Gli/Hh signaling agonist (SAG). Additionally, diet supplementation with *S. frutescens* showed suppression of the formation of poorly differentiated carcinoma in prostates of TRAMP mice [166].

#### 3.1.7. Lamiaceae Plants

Estrogen receptor positive PC-3 cells are more sensitive to the cytotoxic effects of *Nepeta cataria* in comparison with low hormone-receptor presenting DU145 cells. Among multiple extracts and essential oils, only the ethyl acetate extract could expressively decline cell viability in PC-3 cells, in a concentration-dependent way (IC_50_ = 149.6 μg/mL) and showed a sub-G1 peak that showed apoptotic process in ethyl acetate extract-induced cells through enhancement of the expression level of Bax protein and cleavage of caspase-3 and PARP to active [149]. The aqueous extracts of whole plants of *Mentha arvensis*, *M. spicata* and *M. viridis* at a concentration of 100 μg/mL indicated growth inhibition of 75, 85, and 71% respectively against the PC-3 human cancer cell line. The results indicate that *Mentha* spp. have compounds with cytotoxic properties which may find use in developing anticancer agents [148]. *Salvia multicaulis* ethanol extract exhibited antiproliferative activity against both PC-3 and Du145 cells, with IC_50_ values of 240 and 143 µg/mL, respectively [111], while *Salvia triloba* methanolic extract (STE) induced selective cytotoxicity and apoptosis in a concentration-dependent manner, reduced cell motility in the same cells and significantly decreased ANG, ENA-78, bFGF, EGF, IGF-1 and VEGF-D levels in STE-treated DU145 cells, while ANG, IL-8, LEP, RANTES, TIMP-1, TIMP-2, and VEGF levels were considerably reduced in PC-3 cells [164].

#### 3.1.8. Malvaceae Plants

*Helicteres hirsuta* is an herbal medicine used in the management of diabetes and malaria. Leaf and stem extracts and its two sub-fractions (aqueous and saponin-enriched butanol fractions) displayed strong anticancer activity in vitro towards prostate cancer cell lines (IC_50_ = 1.57–152 µg/mL) [142]. *Hibiscus sabdariffa* leaf extract (HLE) dose-dependently inhibited proliferation of LNCaP cells with a 50% growth inhibition value of about 3.0 mg/mL after 24 h. HLE also caused apoptosis and repressed the migration and invasion of human prostate cancer LNCaP cells by inhibiting the activity and expressions of matrix MMP-9 causing NF-κB inactivation mediated via inhibition of the protein kinase B (Akt)/NF-κB/MMP-9 cascade pathway. The inhibitory effect of 50 mg/mL HLE solution in a 5 g diet was confirmed by the inhibition of the growth of LNCaP cells and of the expressions of metastasis-related molecular proteins in vivo in xenograft tumor mice [143].

#### 3.1.9. Phyllanthaceae Plants

*Phyllanthus* plants (*P. amarus*, *P. niruri*, *P. urinaria,* and *P. watsonii)* have antiproliferative activity and induction of apoptosis on prostate cancer cell lines. *Phyllanthus* extracts showed to have notably inhibited the cell adhesion, migration, invasion, and transendothelial migration of PC-3 cells in a dose-dependent way. Additionally, low cytotoxicity on HUVECs was exhibited, suggesting a potential to inhibit tumor metastasis and angiogenesis through the suppression of metalloproteinase enzymes [154].

#### 3.1.10. Poaceae Plants

*Cymbopogon citratus* essential oil consists mainly in limonene (19.33%), *cis*-mentha-1(7),8-dien-2-ol (17.34%), *trans*-mentha-1(7),8-dien-2-ol (13.95%), *trans-para*-mentha-2,8-diene-1-ol (13.91%) and c*is-para*-mentha-2,8-diene-1-ol (8.10%), while *Cymbopogon giganteus* mainly contains geranial/citral A (48.18%) and neral/citral B (34.37%). *C. citratus* essential oil has shown to be most effective on prostate cell lines LNCaP (IC_50_ = 6.36 μg/mL) and PC-3 (IC_50_ = 32.1 μg/mL) vs. 160.1 µg/mL and 303.3 µg/mL for *C. giganteus*. Combining both oils, antagonist, additive, indifferent and synergistic effects were noticed on LNCaP and PC-3 cell lines [135]. *Oryza sativa*, Sung Yod rice is a red-violet pigmented rice produced by an expression method, demonstrated to have more benefits related to cytotoxicity against prostate cancer cells (PC-3) (IC_50_ = 52.06 μg/mL) than the one produced by Soxhlet method extracted with hexane [151].

#### 3.1.11. Rutaceae Plants

Roots of *Fagara zanthoxyloides* and *Pseudocedrela kotchyii* showed activity on PC-3, DU145, LNCaP, and CWR-22 cells [137]. LNCap cells were the most sensitive to the two extracts, with the highest inhibition at day 3 and displaying the maximum rate of apoptosis. IC_50_ values of 5-day cultures showed that *Pseudocedrela* extract had a lower IC_50_ for PC-3, DU145 and LNCap cells (12–20 μg/mL), indicating that these cells were more sensitive to the extract than to the *Fagara* extract, which had significantly higher IC_50_ (25–44 μg/mL) supporting the existence of a potential source of chemopreventive agents for prostate cancer therapy [137]. *Paramignya trimera* (Xao tam phan) has been applied in cancer therapies and cancer-like aliments. Xao tam phan leaf methanol extract (PTL) contains gallic acid, rutin, ellagic acid, protocatechuic acid, and quercetin which has shown to have a pronounced anti-proliferative activity on cancer cell lines including DU145 (growth inhibition of 100% at 100 µg/mL; IC_50_ = 52 µg/mL) [153]. The *Zanthoxyli fructus* water extract demonstrated to have anti-tumor activity against prostate cancer cells LNCaP, DU145, and PC-3. Notably, *Z. fructus* reduced the proliferation of LNCaP and DU145 cells in xenografts in BALB/c nude mice without adverse effects. Additional studies on LNCaP cells exposed that *Z. fructus* blocked AR signaling in conjunction with down-regulation of nuclear levels of AR and decreased the level of the AR-target molecule. It also stimulated apoptosis, prostate-specific antigen, inhibited AKT kinase and down-regulated levels of cyclin D1 protein [173].

#### 3.1.12. Solanaceae Plants

*Capsicum chinense* plant has indicated the cytotoxic effect on LNCaP with percent inhibition greater than 50% at 25 μg/mL [123]. WCE caused elevation of malonyldialdehyde levels, suppression of total antioxidant capacity levels, and increased proliferating cell nuclear antigen expression in the prostate gland in testosterone-induced BPH in rats [171].

#### 3.1.13. Zingiberaceae Plants

The aqueous, ethanol and methanol extracts of turmeric (*Curcuma longa*) presented an antiproliferative effect on PC-3 cells with respectively 34%, 55%, and 60% inhibition [134]. *Zingiber officinale* (ginger), whole extract (GE) has a high growth-inhibitory and death-inductory effects in a wide-spectrum of prostate cancer cells (IC_50_ = 75–512 μg/mL), by impairing reproductive capacity, cell-cycle progression, mitochondrially mediated apoptosis, modulating cell-cycle/apoptosis regulatory molecules. Indeed, daily intake GE (100 mg/kg) repressed about 56% the growth and progression of PC-3 xenografts in nude mice, cut proliferation index and widespread apoptosis matched with controls [174]. Besides, GE combination with its major constituents mainly, 6-gingerol) significantly enhances the antiproliferative effect of GE [175].

#### 3.1.14. Other Plants

The major secondary metabolites of dried stigmas of *Crocus sativus* (saffron) comprise safranal, crocin, and picrocrocin [176]. Saffron has medicinal activities such as apoptosis, suppressing the expression of matrix metalloproteinase, stopping cell cycle progression, modulatory effects on some phase II detoxifying enzymes, and decreasing expression of inflammatory molecules [176]. Saffron extract (SE) showed dose and time-dependent antiproliferative effect on 5 different malignant prostate cancer cell lines (IC_50_ = 0.4–4 mg/mL), by arresting cells at G0/G1 phase and caused apoptosis through strikingly downregulation of Bcl-2 expression, activation of caspase-9 [132]. Xenografted in male nude mice induced with two aggressive prostate cancer cell lines (PC-3 and 22rv1) treated by oral gavage with SE confirmed the in vitro antitumor effects with the mass tumor reduction of 18% in the PC-3 [133]. *Haplophyllum perforatum* and *Urtica dioica* ethanol extract showed antiproliferative activity (IC_50_ = 226 µg/mL on PC-3 cells; IC_50_ = 37 µg/mL on DU145 cells) [111]. Similarly, *Byrsonima crassifolia* plant revealed anti-proliferative activity on LNCaP with more than 50% inhibition percentage at 25 μg/mL [123]. *Nigella sativa* contains a wide range of chemical compounds including thymoquinone (up to 50%), pinene (up to 15%), *p*-cymene (40%), thymohydroquinone, thymol, and dithymoquinone. The molecular mechanisms behind its anticancer activity comprise cell cycle arrest, apoptosis pathways, ROS generation, anti-metastasis/anti-angiogenesis effects, and apoptosis induction, [150].

Several spices have been used to prevent and to treat prostate cancer, including *Allium sativum* (garlic), *N. sativa* (black cumin), and *Piper nigrum* (black pepper). These plants contain numerous essential bioactive compounds, such as thymoquinone, and piperine, which induce apoptosis, inhibit proliferation, migration, and invasion of tumors, and sensitize tumors to radiotherapy and chemotherapy [131].

Polysaccharides from *Psidium guajava* (guava) seed, *Fagopyrum tataricum* (bitter buckwheat), *Fagopyrum esculentum* (common buckwheat), red *Formosa lambsquarters*, and yellow *F. lambsquarters* indirectly significantly inhibit PC-3 cell growth by immunotherapy with splenocyte- and macrophage-conditioned media (SCM or MCM) presenting a negative link between PC-3 cell viabilities and IL-6 + TNF-α)/IL-10 level ratios in MCM proposing that macrophages suppress PC-3 cell growth through diminishing secretion ratios of proinflammatory/anti-inflammatory cytokines in a tumor microenvironment [138]. The aqueous extract of *P. guajava* (PE) budding leaves exhibited cytotoxicity with the IC_50_ of ~0.57 mg/mL on DU145 cells and effectively hindered the expressions of VEGF, IL-6 and IL-8 cytokines, and MMP-2 and MMP-9, and simultaneously activated TIMP-2 and suppressed the cell migration and the angiogenesis [157]. At 1.0 mg/mL, PE reduced the viability of DU145 cells to 36.1 and 3.59%, respectively, after 48 h and 72 h and lowered the colony forming capability of DU145 cells [177]. This antiproliferative activity was conserved either in the presence or the absence of synthetic androgen R1881 in LNCaP cell and arrest cell cycle at G(0)/G(1) phase with a high quantity of apoptotic LNCaP cells. The treatment with PE considerably reduced both the PSA serum levels and tumor size in a xenograft mouse tumor model [158].

The anticancer potential of *Aloe perryi* flowers showed the percentage inhibition of various extracts (viz. petroleum ether, chloroform, ethyl acetate, butanol and aqueous) on PC-3 of 88.9% for petroleum ether extract [114]. *Urtica dioica* extract expressively repressed the cell growth with 24h and 48 h (IC_50_ = 29.46 and 15.54 µg/mL, respectively), being able to induce apoptosis in PC-3 cells by substantially increasing the caspase-3 and 9 mRNA expression, while decreasing Bcl-2, and arrested the cell cycle in G2 stage [167]. Extracts from leaves and stems of *Chenopodium hybridum* containing mainly rutin (2.80 μg/g dry weight), 3-kaempferol rutinoside (2.91 μg/g), 4-OH-benzoic (1.86 μg/g) and syringic acids (2.31 μg/g) showed a low cytotoxic activity except for the extract from the leaves, which was effective against prostate Du145 cell line with 98.28% dead cells at 100 μg/mL [127].

Pomegranate (*Punica granatum*) fruit as well as its juice, extract, and oil have anti-proliferative, and anti-tumorigenic properties by modulating multiple signaling pathways, suggesting its use as a promising chemopreventive/chemotherapeutic agent of prostate cancer [5]. A peel extract of pomegranate fruit exhibited antiproliferative properties on PC-3 cells, reducing the cell viability to values below 40% (IC_50_ < 5 μg/mL) and maximum mean growth inhibition of 79.3% [178]. Pomegranate seed methanolic extract reduced the cell viability to values below 23%, even at the lowest doses, and a maximum means growth inhibition of 81.4% in the same cells [161]. Albrecht et al. [159] demonstrated significant antitumor activity of pomegranate cold-pressed (oil) or supercritical CO_2_-extracted (S) seed oil, fermented juice polyphenols (W), and pericarp polyphenols (P) on LNCaP, PC-3, and DU145 human prostate cancer cells (IC_50_ = 70 µg/mL), whereas normal prostate epithelial cells (hPrEC) were expressively less affected (IC_50_ = 250 g/mL). Modifications in both cell cycle distribution and induction of apoptosis were noted. All extracts potently suppressed PC-3 invasion through Matrigel, and furthermore, P and S demonstrated potent inhibition of PC-3 xenograft growth in athymic mice [159].

Polysaccharide extracted from *Polygonatum* spp. selectively restrained the growth of prostate-cancer-associated broblasts (CAFs) by stimulating autophagy in prostate-CAFs via the activation of Beclin-1 and LC3 (key autophagy proteins) and thus developing the effectiveness of cancer therapy [156]. *Costus pulverulentus* ethanol extract of the stem (campesterol, stigmasterol β-sitosterol, vanillic acid, among other components) presented moderate cytotoxic effects on PC-3 cells (IC_50_ = 179 µg/mL) and induced DNA damage in the comet assay at 200 µg/mL or higher concentrations [129]. The study of *Ficus deltoidea* var. *angustifolia* (FD1) and var. *deltoidea* (FD2) including crude methanolic extracts, *n*-hexane (FD1h, FD2h), chloroform (FD1c, FD2c) and aqueous extracts (FD1a, FD2a) fractions on prostate cancer cells revealed that FD1c and FD2c are the most cytotoxic extracts against both prostate cancer cell lines (IC_50_ = 23 and 29 µg/mL, respectively, for PC-3 and 19 and 23 µg/mL, respectively, for LNCaP). This occurs through induction of cell death via apoptosis as supported by nuclear DNA fragmentation and increased metalloproteinase depolarization, activation of caspase-3 and -7, and inhibited both migration and invasion of PC-3 cells and down-regulated Bcl-2, VEGF-A and CXCL-12 gene expressions, while that of Bax and Smac/DIABLO were up-regulated. The LC-MS dereplication identified isovitexin in FD1c; and oleanolic acid, moretenol, betulin, lupenone, and lupeol in FD2c [139].

Zhang et al. [152] explored whether total glucosides of paeony, extracted from the root of *Paeonia lactiflora*, suppressed lipopolysaccharide (LPS)-stimulated proliferation, migration, and invasion in androgen-insensitive prostate cancer cells. At 312.5 μg/mL, these compounds suppressed LPS-stimulated proliferation of PC-3 cells, inhibited activation of NF-κB and mitogen-activated protein kinase p38 in LPS-stimulated PC-3 cells and inhibited inflammation-associated STAT3 activation and proliferation, migration and invasion in PC-3 androgen-insensitive prostate cancer cells [152]. *Melissa officinalis* hydroalcoholic extract presented a high potency to inhibit proliferation of prostate cancer cells in a dose-independent manner with the mean growth inhibition of 79.9% (maximum growth inhibition of 83.7% at a dose of 500 µg/mL, minimum growth inhibition of 76.29% at 20 µg/mL, respectively. and IC_50_ < 500 µg/mL on PC-3 cells) [147], by inhibiting the expression of p53, Bcl-2, Her2, VEGF-A and hTERT in human prostate cancer cell lines [146].

*Carica papaya* (papaya) leaf juice (LJP) extract revealed selective anti-proliferative and anti-metastatic effect against cell lines representing benign hyperplasia, tumorigenic and normal cells of prostate origin prostatic diseases, including prostate cancer. LJP displayed potent antiproliferative effects after 72 h treatment on RWPE-1, BPH-1, PC-3 and LNCaP cells (IC_50_ = 220, 790, 950 and 960 µg/mL, respectively). Comparable to LJP, in vitro digested LJP also showed an antiproliferative effect after 72 h (IC_50_ = 1460, 1320, 2270 and 4240 µg/mL on RWPE-1, BPH-1, PC-3, and LNCaP cells, respectively). The medium polar fraction of LJP showed broad-spectrum efficacy (similar to paclitaxel) and selective anti-proliferative activity (IC_50_ = 20–70 µg/mL) against cells with several phases of prostatic diseases, induced cell cycle arrest at S phase and apoptosis, and notably inhibited migration and adhesion of PC-3 cells [125]. Likewise, *A. wallichii* (aqueous ethanol extract) containing flavonoids, steroids, terpenoids, reducing sugars and glycosides exhibited antiproliferative activity on the PC-3 cell line (IC_50_ = 69.69 μg/mL) [113].

Hypoxia enhances cancer development in solid tumors. HIF-1α is a transcription factor, expressed under hypoxia in solid tumor cells, that regulates several target genes. It is involved in cancer progression, including angiogenesis, metastasis, anti-apoptosis, cell proliferation and is related to resistance to cancer treatment [130]. *Crataegus Pinnatifida* Schneider ethanol extract decreased cell growth, HIF-1α and sphingosine kinase-1 (SPHK-1) in hypoxia-induced human prostate cancer DU145 cells [130].

C4-2 prostate cancer cell treatment with muscadine grape skin extract (MSKE) from muscadine grape (*Vitis rotundifolia*) which is a common red grape used to produce red wine, indicated that this extract could unfold protein response that can eventually lead to apoptosis in prostate cancer cells [168]. The saponin fraction isolated from *Lysimachia ciliata* (CIL-1/2) exposed no cytotoxic or cytostatic effect at the concentration of 0.5 µg/mL prostate cancer cell lines (DU145, PC-3). In contrast, cocktails of CIL-1/2 and mitoxantrone (a drug commonly used in prostate cancer therapy) displayed synergistic cytostatic and proapoptotic effects on prostate cancer cell movement and invasiveness [144]. Besides, the hydroalcoholic extract of *Cornus mas* fruit reduced PC-3 cell viability below 26%, even at the lowest doses, with a mean growth inhibition of 81.6% [128].

A *Leucas aspera* nanoformulation showed a concentration- and time-dependent in vitro cytotoxicity on PC-3 cells that was confirmed by the in vitro hemolysis assay, cellular uptake studies, cell aggregation studies, and cell migration assays [179]. *Moringa oleiferna* inhibited the cell proliferation of PC-3 cell lines in a concentration-dependent manner (IC_50_ = 14.3 and 22.2 μg/mL) [110]. Silver nanoparticles (AgNps) of *Guiera senegalensis* leaves extract presented antiproliferation effect on PC-3 cell lines in a concentration-dependent manner (IC_50_ = 23.48 μg/mL), indicating its potential applications in pharmacology for the treatment of cancers [180]. Similarly, AgNps from the aqueous extract of *H. thebaica* fruit inhibited the proliferation of prostate cancer cell lines in a dose dependent manner (IC_50_ = 2.6 mg/mL) [181]. The methanol extract of *Maytenus royleana* (MEM) leaves and its fractions showed to possess antiproliferative effect on prostate cancer cells. The effect was supplemented by G2 phase arrest of cell cycle, increase in cdk inhibitors and downregulation of cyclin/cdk network. MEM treated cells showed cleavage of caspase-3 and PARP, and modulation of apoptotic proteins, determining apoptosis as the primary mechanism of cell death. Remarkably MEM suppressed AR/PSA signaling both in prostate cancer cell cultures and in the in vivo model. Intraperitoneal injection of MEM (1.25 and 2.5 mg/animal) to athymic nude mice implanted with androgen-sensitive CWR22Rν1 cells presented major inhibition in tumor growth and decreased serum PSA levels [145]. *Remotiflori radix* ethanol extract (ERR) at >100 μg/mL caused dose- and time-dependent cell death in PC-3 and DU145 cell lines (autophagic and apoptotic effects, induction of LC3 punctuation, YO-PRO-1 uptake, DNA fragmentation, activation of caspases, and PARP cleavage). Phosphorylation of AMPK, ULK, and p38 was increased after ERR treatment, and CHOP (ER stress marker) was also raised. Additionally, oral administration of ERR at 50 mg/kg efficiently repressed the tumorigenic growth of PC-3 cells with no adverse effects [163].

*Berberis libanotica* Ehrenb (BLE) is a plant rich in alkaloids with high potential for eliminating advanced prostate cancer cells involved in tumor resistance toward conventional tumor therapy. *B. libanotica* indicated a dose- and time-dependent manner cytotoxicity and cell death mainly at high concentrations (60 and 100 mg/mL) of DU145, PC-3 and 22Rv1 cells, by perturbating the cell cycle, leading to a G0-G1 arrest, Ros production, and also inhibited cell migration and invasion, suggesting a role in inhibiting metastasis [122]. The oral administration of AGN root ethanol extract showed chemopreventive effects against the growth of prostate epithelium and neuroendocrine carcinomas (NE-Ca) in the TRAMP model [116]. *Eurycoma longifolia* standardized total quassinoids composition (SQ40) with 40% of total quassinoids repressed LNCaP cell growth and RWPE-1 human prostate normal cells (IC_50_ = 5.97 and 59.26 μg/mL, respectively). SQ40 also inhibited 5α-dihydrotestosterone-stimulated growth in LNCaP cells dose-dependently, anchorage-independent growth, suppressed LNCaP cell growth via G0/G1 phase arrest, down-regulated CDK4, CDK2, Cyclin D1, and Cyclin D3, and up-regulated p21Waf1/Cip1 protein levels. At higher concentrations or longer treatment duration, SQ40 can lead to G2M growth arrest and apoptotic cell death (PARP cleavage) in LNCaP cells. Additionally, SQ40 can also inhibit AR translocation to nucleus which is critical for the transactivation of its target gene, PSA, resulting in a substantial decrease of PSA secretion after the treatment. Finally, intraperitoneal injection of 5 and 10 mg/kg of SQ40 showed to considerably suppress the LNCaP tumor growth on mouse xenograft model [136].

### 3.2. Plant-Derived Bioactives with Anti-Prostate Cancer Potential

Many classes of metabolites isolated from medicinal plants have been reported for their activity against prostate cancer, namely alkaloids, phenolic compounds, and terpenoids (Table 4).

#### 3.2.1. Alkaloids

6-Hydroxycrinamine, lycorine, and crinamine three alkaloids from *Crinum asiaticum* showed potent hedgehog (Hh)/GLI1-mediated transcriptional inhibitory activity (IC_50_ = 14.3, 19.4, and 4.7 µM, respectively) and revealed cytotoxicity against DU145 cells both PANC1 cells (IC_50_ = 22.7, 19.9, and 10.0 µM, respectively) and DU145 cells (IC_50_ = 9.3, 21.7, and 18.5 µM, respectively), without affecting normal cell lines. These compounds repressed the Hh signaling pathway by down-regulating the expression of GLI-related proteins (PTCH and BCL2) in DU145 cells [183]. Likewise, liriodenine, (−)-anonaine, (−)-caaverine, (−)-nuciferine, 7-hydroxydehydronuciferine from the leaves of *Nelumbo nucifera* Gaertn. cv. rosa-plena have shown significant cytotoxicity in DU145 cells (IC_50_ = 95.4, 150.1, 94.4, 218.4, and 80.8 μM, respectively) [182]. Capsaicin, the spicy ingredient of red hot chili peppers, has been shown to induce death in LNCaP and PC-3 in a time- and concentration-dependent manner by increasing the levels of microtubule-associated protein light chain 3-II (a marker of autophagy) and the accumulation of the cargo protein p62, indicating an autophagy blockage and triggered ROS generation in cells [184]. Emetine, a naturally-derived alkaloid from the *Carapichea ipecacuanha* plant, has been shown to have the potential for anti-tumorigenic effects for cancer treatments. Novel emetine dithiocarbamate (EMTDTC) analogs EMTDTC-55 and EMTDTC-56 shown important anti-tumorigenic activities on DU145, LNCaP and PC-3 and apoptotic potential effect in the prostate cancer cells. These compounds have also shown major chemotherapeutic potential in moderately metastatic DU145 and highly metastatic PC-3 cells [185]. Lycorine (an alkaloid extracted from Amaryllidaceae family plants) has been reported to inhibit proliferation, induced cell apoptosis, and cell death in various prostate cancer cell lines. It has also withdrawn both growth and metastasis in multiple organs in in vivo assays and improved mice survival in subcutaneous and orthotopic xenotransplantations, by ectopic implantation of the human hormone-refractory PC-3M-luc cells. Additionally, lycorine showed to block EGF-induced JAK/STAT signaling and induce STAT expression [187]. Sophocarpine, matrine, oxymatrine, and oxysophocarpine from radix *Sophorae flavescentis,* a plant used for the treatment of different stages of prostate cancer in China, has demonstrated antiproliferative and apoptosis activity. Specifically, matrine had good inhibitory effects, with half maximal inhibitory concentration values of 0.893 mg/mL [188]. Schisanspheninal A from the hexane extract of fruits of *Schisandra grandiflora* revealed anti-proliferative activity against DU145 (IC_50_ = 165 µM) [189]. Tetrandrine, isolated from traditional Chinese medicinal plant *Stephania tetrandra,* has also reported antiproliferative activities against PC-3 (IC_50_ = 7.68 μM) [190].

#### 3.2.2. Phenolic Compounds

##### Flavonoids

Treatment of low-metastatic LNCaP and high-metastatic PC-3 cells with green tea polyphenols (-)-epicatechin-3-*O*-gallate and (-)-5,7-difluoro-epicatechin-3-*O*-gallate confirmed a dose-dependent inhibition of cell viability. The effects propose that (-)-epicatechin-3-*O*-gallate and the more effective (-)-5,7-difluoroepicatechin-3-*O*-gallate might be therapeutically used to inhibit tumorigenesis during initiation, promotion, and progression by reducing inflammation [193]. Polyphenolic compounds 7-O-galloyl catechin (GC), catechin (C), methyl gallate (MG), and catechin-3-O-gallate (CG) from *Acacia hydaspica* caused cell death of PC-3 cell, in a dose-dependent manner via apoptosis. The treatment suppressed the expression of Bcl-2, Bcl-XL and survivin, and down-regulated the signaling pathways of AKT, NF-κB, ERK1/2 and JAK/STAT in the same cells [195]. Licoricidin, which inhibits metastasis and isoangustone A, which induces apoptosis G1 cycle arrest, are two active components found in glycyrrhizin hexane/ethanol extract of *G. uralensis.* Their activity was related to reduced activation of proteases, and the levels of adhesion molecules may constitute a component of the mechanism [221,222]. Moreover, *Hesperetin*, a flavonoid found in numerous citrus species, has been reported to have an antiproliferative effect in PC-3 cells and to cause the rise of IL-6 gene expression, IL-6 protein secretion, signal transducer and activator of transcription 3 (pSTAT3), extracellular signal-regulated kinases 1/2 (pERK1/2) and pAKT signaling pathways. Likewise, hesperetin treatment induced cell cycle arrest at the G1 phase and can be considered as an agent to synchronize and stop cell cycle at G0/G1 phase [216].

Over expressed MiR-21 in prostate cancer is associated with metastasis and drug resistance to chemotherapy with DX. 4′,5,7-Trihydroxy-3′,5′-dimethoxyflavone (tricin) isolated from *Allium atroviolaceum* has potentiated the effect of DX on PC-3 cell proliferation (IC_50_ = 117.5 μM and 0.1 nM for tricin and DX, respectively). The synergistic effect of the combination of Tricin and DX significantly decreased the proliferation of PC-3 cells. Besides, MiR-21 in treated cells with Tricin pointedly reduced compared with the control, suggesting the aptitude of tricin to efficiently decrease metastasis and drug resistance of DX [243]. A flavonoid of the flavanone class, HLBT-100 (or HLBT-001), isolated from *Tillandsia recurvata* exhibited potent anti-prostate cancer (IC_50_ values < 0.100 µM), affecting the cell cycle, activate caspase-3/7, cause DNA fragmentation culminating in apoptotic cell death. It has also shown antiangiogenic potential by inhibiting capillary sprout and tube formation in a dose-dependent manner in the ex vivo rat aortic ring [218]. Altholactone, another natural flavanone isolated from *Goniothalamus* spp., has confirmed anticancer activity against DU145 by stimulating cell cycle arrest in S phase and triggered apoptosis, and inhibition of NF-κB and STAT3 activity [197]. In another study, 6-prenylnaringenin and 8-prenylnaringenin, two prenylflavonoids present in *Humulus lupulus* displayed a dose-dependent reduction of cellular proliferation of PC-3 cells [194]. Isovitexin from *Ficus deltoidea* has also shown antiproliferative activity against PC-3 cells (IC_50_ = 43 μg/mL) [139].

Apigenin, a naturally occurring plant flavone, has been shown to have anti-proliferative, and anti-carcinogenic activities. Orally administrating apigenin to TRAMP mice (20 and 50 μg/mouse/day, 6 days/week for 20 weeks), resulted in a reduction in tumor volume of the prostate and a stop in metastasis, which was linked with inhibition of NF-κB activation and binding to the DNA. This flavone led to apoptosis via downregulation of the expression of NF-κB-regulated gene products related to proliferation (cyclin D1, and COX-2), anti-apoptosis (Bcl-2 and Bcl-xL), and angiogenesis (VEGF) and increased cleaved caspase-3 labeling index in the dorsolateral prostate [198]. Another natural flavone found in several plant extracts chrysin, induced death of prostate cancer cells by apoptosis supported by DNA fragmentation and increasing the population of both DU145 and PC-3 death cells in the sub-G_1_ phase of the cell cycle (inducing mitochondrial-mediated apoptosis and ER stress, and regulating signaling pathways responsible for proliferation of prostate cancer cells) [199]. Maysin isolated from *Zea mays* corn silks dose-dependently reduced the PC-3 cell viability, with an 87% reduction at 200 μg/mL via stimulation of mitochondrial-dependent apoptotic cell death as indorsed by DNA fragmentation, depolarization of mitochondrial membrane potential, and reduction in Bcl-2 and pro-caspase-3 expression levels. This compound also significantly diminished the phosphorylation of Akt and ERK. Combined treatment with maysin and other anticancer agents, including 5-FU, etoposide, cisplatin, or camptothecin, synergistically increased PC-3 cell death [227]. A compound obtained from the leaves of *G. biloba,* ginkgetin, has demonstrated to inhibit both inducible and constitutively activated STAT3 and to block the nuclear translocation of p-STAT3 in DU145 prostate cancer cells [215]. It has selectively constrained the growth of prostate tumor cells by inducing STAT3 dephosphorylation at Try705 and inhibited its localization to the nucleus, leading to the inhibition of expression of STAT3 target genes such as cell survival-related genes (cyclin D1 and survivin) and anti-apoptotic proteins (Bcl-2 and Bcl-xL) and thereby inhibiting the growth of STAT3-activated tumor cells. The inhibition of tumor growth in xenografted nude mice and downregulated p-STAT3(Tyr705) and survivin in tumor tissues was also reported in this study [215].

Afzelin, a flavonol glycoside that was earlier isolated from *Nymphaea odorata* inhibited the proliferation of LNCaP and PC-3 cells through inhibition of LIM domain kinase 1 expression and blocked the cell cycle in the G_0_ phase [196]. Likewise, rutin, another flavonol glycoside isolated from *Solanum macaonense* releaved to have selective moderate cytotoxicity toward DU145 cancer cell lines (IC_50_ = 31.8 μM) [241]. A flavonoid glycoside from *Epimedium* herb extract, icarisid II, indicated a very potent antiproliferative effect on LNCaP cells at 10 µM with less than 25% of cell viability and suppressed the expression of the androgen-responsive KLK3 gene [220].

Quercetin, a flavonoid compound ubiquitous in numerous dietary plants, possesses evidenced potential in treating advanced metastatic castration-resistant prostate cancer [235,236]. Quercetin not only resulted in a rise in the G2/M phase population in both PC-3 and LNCaP cells but also increased the S phase population in PC-3 cells (IC_50_ = 22.12 μM for PC-3 and 23.29 μM for LNCaP cells) [237]. Quercetin and its derivatives anti-proliferative activities towards both androgen-refractory and androgen-sensitive prostate cancer cells signpost that 3,4′,7-O-trialkylquercetins were much more potent than quercetin towards prostate cancer cells [235]. Fisetin is a plant flavonoid with therapeutic potential against such as prostate cancer. Metabolomic analysis of tumor xenografts from fisetin-treated animals recognized several metabolic targets with hyaluronan (HA) as the most affected. Fisetin showed to downregulate secreted and intracellular HA levels on both in vitro and in vivo studies model of prostate cancer [210]. Flavopiridol, a flavonoid alkaloid, showed high antiproliferative potential at a very low concentration of 0.1 ng/mL in three of four prostatic xenografts. Overall, in 61% tumor xenografts the drug treatment resulted in an IC70 of <10 ng/mL. Later, in vivo studies corroborated this antitumor activity in prostate cancer xenografts investigated at a maximally tolerated dose (10 mg/kg/day, administered orally), causing tumor regression in PRXF1337 and tumor stasis lasting for 4 weeks in PRXF1369 [212].

##### Anthocyanidins

Delphinidin is a major anthocyanidin compound found in many fruits that considerably inhibit prostate cancer cells cell growth, but with a different dose response of prostate cancer cells to this drug (IC_50_ = 50, 70, 65, and 90 μM for LNCaP, C4-2, 22Rν1, and PC-3 cells, respectively, 48 h post-treatment). Delphinidin treatment prevents the tumorigenic potential of PC-3 cells in in vivo preclinical setting via interference with the NF-κB signaling pathway [207]. This anthocyanidin induces p53-mediated apoptosis by suppressing histone deacetylase activity through increased caspase-3, -7, and -8 activity and activating p53 acetylation in LNCaP cells [206].

##### Phenols

The dehydrozingerone from the rhizome of *Z. officinale* and its derivatives displayed potent cytotoxic potential (IC_50_ = 1.8–3.0 μM on PC-3 cell lines), and the derivatives have shown to be more potent than the parent compound [205]. Cinnamaldehyde and eugenol are naturally present in cinnamon, bay leaf and eugenol is abundantly present in clove. They both possess 64–75% cytotoxic activity against PC-3 cell line at 100 µM concentration [200]. Garcinol, a polyisoprenylated benzophenone isolated from *Garcinia indica* fruit rinds of revealed antitumor activity by inducing apoptosis and inhibiting autophagy in human prostate cancer cells at 30 µM. Also, in vivo assays showed the reduction of the tumor size more than 80% after the mouse treatment, confirming the in vitro results [213]. Garcinol showed to inhibit several key regulatory pathways (e.g., NF-κB and STAT3) in cancer cells, thus explaining its ability to control the malignant growth of solid tumors in vivo [214]. This drug is still in the preclinical stage due to a lack of systematic and conclusive evaluation of pharmacological parameters [214]. A phenol from the root bark of *Paeonia moutan* and the grass of radix *Cynanchi Paniculati*, paeonol, has been reported to block growth of prostate cancer cells, DU145 and PC-3 in dose-and time-dependent manner, induce apoptosis and enhance activities of caspase-3, caspase-8, and caspase-9, reduced expression of Bcl-2, and rise expression of Bax, cut phosphorylated status of Akt and mTOR in DU145. Paeonol and PI3K/Akt inhibitor exhibited a synergistic antiproliferative effect on DU145 cells. Furthermore, the oral administration of paeonol to the DU145 tumor-bearing mice considerably lowered tumor cell proliferation and led to tumor regression [230]. Resveratrol, another phenol, has demonstrated to inhibit cell growth induced by androgen R1881 or 17β-estradiol in a concentration-dependent manner at concentrations as low as 1 μM for androgen and as low as 5 μM for 17β-estradiol. In a xenograft model, resveratrol hindered LNCaP tumor growth and repressed expression of a marker for steroid hormone responses [239]. Amongst tocotrienols, δ-tocotrienol has exhibited the highest anti-cancer activity. The combination of δ-tocotrienol and γ-tocopherol (10 μM + 5 μM, respectively) showed synergy in anti-prostate cancer activity against LNCaP by stopping both cell cycles in the G1 and G2/M phases [192]. A phenol glycoside from stems and leaves of *Physalis angulata*, physangulatins I, indicated antiproliferative effects against human prostate cancer cells (IC_50_ = 1.17 and 2.12 μM in C4-2B and 22Rvl cells, respectively), and inhibitory effects on NO production induced by LPS in macrophages (IC_50_ = 3.51 μM) [232]. Hirsutenone, from *Alnus japonica*, can directly inhibit of Akt1/2 and suppress anchorage-dependent and independent cell growth, inducing apoptosis in both PC-3 and LNCaP cells, by direct binding in adenosine triphosphate (ATP)-noncompetitive manner. These results have also been showed ex vivo [217]. Additionally, hirsutenone treatment has proven to reduce phosphorylation of mTOR, a downstream substrate of Akt, without disturbing Akt phosphorylation [217].

##### Lignans

Magnolol is a lignan found in the roots and bark of *Magnolia officinalis* (magnolia tree). After a 24-h exposure (40 and 80 μM), presented cytotoxicity with 49–50% and 50–76% of inhibition, respectively, affecting cell cycle progression of DU145 and PC-3 cells. This induced shifts in the cell cycle and a reduction in the proportion of cells entering the G2/M-phase. Magnolol repressed the expression of regulatory proteins (e.g., A, B1, D1, E, CDK2, and CDK4 cyclins). While this lignin led to a decrease in retinoblastoma protein (pRb) pRBp107, but pRBp130 increased. Protein expression levels of p16(INK4a), p21, and p27 apparently did not change after a 24 h exposure, however, after exposure at 6 h did increase p27 protein expression levels, demonstrating that magnolol can modify the performance of androgen-insensitive DU145 and PC-3 cells [224]. A norlignan from *Peperomia tetraphylla,* peperotetraphin, was reported to inhibit the PC-3 cell growth in a dose- and time-dependent fashion and to induce the cell cycle arrest at the G1-S phase, causing apoptosis [231].

##### Naphthoquinones

Plumbagin (5-hydroxy-2-methyl-1,4-naphthoquinone) is a plant-derived naphthoquinone obtained mostly from the families Plumbaginaceae, Droseraceae, and Ebenaceae [261]. Reports have shown that this compound can strongly inhibit the invasion of DU145, PC-3, and CWR22rv1 and induce apoptosis in PC-3, LNCaP, and C4-2 cells (5 μM and 20 μM). The intraperitoneal administration of 2 mg/kg body weight of plumbagin resulted in delayed tumor growth by 3 weeks and reduced both tumor weight and volume by 90% [233]. A stable nanoformulation of plumbagin nanoparticles from *Plumbago zeylanica* root extract in vitro cytotoxicity exhibited concentration and time-dependent toxicity on prostate cancer cells. Nonetheless, the drug was described as highly toxic to normal cells (when compared with plumbagin nanoformulation). Plumbagin nanoparticles were found biocompatible, demonstrating antimetastasis and apoptotic activity in prostate cancer [155,261].

##### Tannins

Cornusiin A, camptothin B, and cornusiin H are tannins from the acetone extract of *Cornus alba*. These compounds have shown selective antiproliferative effects on LNCaP hormone-dependent prostate cancer (IC_50_ = 48.32, 41.48, 44.06 μM, respectively) more potently than DU145 cells (IC_50_ = 6.31, 6.03, 5.97 μM, respectively), inducing apoptosis and S-phase arrest [141]. Ellagic acid has also proved to have anticancer effects against in LNCaP, PC-3, and DU145 (weaker in these two cells lines), by declining cell proliferation through phosphorylated STAT3, ERK, and AKT cellular signaling proteins reduction [209]. It also induces apoptosis by caspase-3 activation in the in vivo TRAP model, increasing Bax/Bcl-2 ratio, and cut the level of lipid peroxidation in ventral prostate [208]. Punicalagin, another polyphenol from pomegranate, has also displayed potent anticancer activity in prostate cancer, with concentration-dependent selective inhibition of viability in PC-3 and LNCaP at 10–100 µM, via stimulation of apoptosis and anti-angiogenic effect [234]. Likewise, tannic acid dependently inhibited the proliferation of PC-3 and LNCaP (IC_50_ = 35.3 μM and 29.1 μM, respectively). This compound worked by significantly inhibit the migration of prostate cancer cells (92.7%) and have an anti-invasive potential of PC-3 cells (80.9%). Tannic acid enlarged early apoptosis rate of PC-3 and LNCaP cells but also regulated protein and mRNA expressions of several enzymes (e.g., CYP17A1, CYP3A4, CYP2B6, NQO1, GSTM1, and GSTP1) [242].

##### Coumarins

Decursin is a coumarin from AGN or KMKKT root ethanolic extract, known as an antiandrogen and AR compound that can suppress PSA expression after 48-h exposure on prostate cancer cells (IC_50_ = 0.4 µg/mL). Decursin has been reported to repeat the neuroendocrine differentiation induction and G1 arrest actions of AGN or KMKKT to inhibit androgen-stimulated AR translocation to the nucleus and down-regulate the AR protein abundance without affecting the AR mRNA level [262]. In an in vivo study, TRAMP mice gavage-treated daily with excipient vehicle, AGN (5 mg/mouse) or equimolar decursin (D)/isomer decursinol angelate (DA) (D/DA) (3 mg/mouse) suggested that D/DA act as probable active/prodrug compounds against epithelial lesions, and might assist with non-pyranocoumarin molecules to entirely express AGN efficacy against NE-Ca. Mice bearing NE-Ca and treated with AGN-and D/DA showed inhibition in prostate growth. Higher survival of mice was improved by AGN, but not by D/DA, yet AGN-and D/DA-treated mice had lower NE-Ca burden [117]. Lastly, osthol, a natural coumarin isolated from Apiaceous plants has shown protective and therapeutic in PC-3 (IC_50_ = 28.81 μM) by regulating apoptosis, proliferation and invasion mediated by multiple signal transduction cascades [4,228].

##### Polyphenols

It has been reported that *Z. officinale* polyphenolic components viz., 6G(6-gingerol), 8G(8-gingerol), 10G (10-gingerol), 6S (6-shogoal) are more active than whole ginger extract against prostate cancer cells (IC_50_ = 22.07 μg/mL for 6G, 3.22 μg/mL for 8G, 17.53 μg/mL for 10G, 1.12 μg/mL for 6S, and 250 μg/mL for ginger extract) [175]. Additionally, chlorogenic acid (CA), a polyphenol from *C. pinnatifida* was studied for the treatment of DU145 cells for 48 h. The outcome reported a reduction in cancer cell growth, phosphorylation AKT and glycogen synthase kinase-3β (GSK-3β), which are related with HIF-1α stabilization and disturbed SPHK-1 in a concentration-dependent manner the secretion and cellular expression of VEGF (preventing hypoxia-induced angiogenesis). This effect was confirmed by a reduction in proliferation of cell nuclear chlorogenic acid antigen [130]. Curcumin (diferuloylmethane), a polyphenol originated from *C. longa* [201], has been shown to have an antitumor effect in several biological pathways implicated in tumorigenesis, mutagenesis, apoptosis, cell cycle regulation and metastasis [204]. Compared with the parent curcumin, curcumin nanoparticles (nanocurcumin) present noteworthy activity against PC-3 cells and low toxicity against normal cells (HEK) [201]. The in vitro and in vivo efficacy of curcumin in the treatment of prostate cancer, especially for castration-resistant prostate cancer, has also been demonstrated [202,203]. *Magnolia* spp. has a drug, honokiol, which can be isolated from bark and leaves. This biphephonol has proven chemopreventive and/or therapeutic effects against prostate cancers, affecting multiple signaling pathways, molecular and cellular targets (e.g., NF-κB, STAT3, EGF receptor, cell survival signaling, cell cycle, cyclooxygenase, and other inflammatory mediators) [263]. Honokiol showed to be highly effective in dose-and time-dependently inhibit the viability of LNCaP and C4-2, cutting the protein level of AR, androgen-stimulating nuclear translocation of AR, and transcriptional activity of AR in prostate cancer cells lines [219].

##### Xanthones

Mangiferin (1,3,6,7-tetrahydroxyxanthone-C2-β-d-glucoside) is a xanthonoid found in plants, for example, *Mangifera indica* [226]. In a study, mangiferin has shown to reduce the proliferation of PC-3 cells in a concentration-dependent manner (20 µM, for 72 h, or 40 µm for 48 or 72 h). This drug led to dose-dependent apoptosis and enhanced the caspase-3 activity in PC-3 cells [225]. A single administration of mangiferin or in combination with recognized anticancer drugs revealed possible benefits. Pharmaceutical development, clinical trials on cancer targets are still lacking [226]. α-Mangostin from *Garcinia mangostana* (mangosteen fruit) has shown biological effects in prostate cancer cells by selectively upregulating ER stress markers and causing apoptosis in 22Rv1 and LNCaP prostate cell lines. In an in vivo assay, this compound has also significantly suppressed tumor growth in mice, without obvious toxicity [191].

##### Chalcones

Oxyfadichalcones A-G isolated from the aerial parts of *Oxytropis chiliophylla* were evaluated and revealed cytotoxic activities against the PC-3 cell line (inhibition: 57.9–97.6%) [229]. Similarly, resveratrol, found in large amounts in the skin of grapes, tomatoes, and in red wine [264] can reduce the proliferation of PC-3 and LNCaP cells (>10–50 μM for 48 h) [238]. *Humulus lupulus* contains xanthohumol, a prenylated chalcone, which has been tested on LNCaP in combination with TRAIL. The drug has demonstrated antiproliferative effect mediated by apoptosis via activation of caspases-3, -8, -9, Bid, rising expression of Bax, and reducing expression of Bcl-XL and mitochondrial membrane potential [244]. Li et al. [211] revealed that flavokawain A (FKA), a chalcone from *Piper methysticum* (kava), selectively constrained the growth of pRb deficient cell lines, resulting in a proteasome-dependent, and ubiquitination-mediated Skp2 degradation. Finally, dietary feeding TRAMP mice with FKA led to clear anti-proliferative and apoptotic effects via down-regulation of Skp2 and NEDD8 and up-regulation of p27/Kip1 in the mice’s prostate [211].

##### Carotenoids

Crocin, *C. sativus* major constituent, has revealed antiproliferative effects (IC_50_ = 0.26–0.95 mM/mL) on prostate cancer cells by arresting cell cycle progression and inducing apoptosis in prostate cancer [132]. Both crocin and crocetin reduced tumor growth in PC-3 and 22rv1 xenografts [133], however, crocetin exhibited superior antitumor effects when compared with crocin and SE with the mass tumor reductions of 38%, and 75%, respectively, in PC-3 xenografts. Both treatments regressed the epithelial-mesenchymal transdifferentiation and repressed prostate cancer cell invasion and migration via downmodulation of metalloproteinase and urokinase expression/activity, which suggests that these drugs may affect metastatic processes [133].

#### 3.2.3. Terpenoids

##### Sesquiterpenes

A sesquiterpene isolated from *Santalum* spp. (sandalwood), α-santalol, has demonstrated anticancer effects, inducing cell-cycle arrest and apoptosis in both in vitro and in vivo models of prostate cancer [248]. Likewise, sesquiterpenes viz. widdarol-peroxide, widdaranal A, widdaranal B and isocuparenal from the hexane extract of *Schisandra grandiflora* fruits, indicate anti-proliferative activity against prostate cancer cells DU145 (IC_50_ = 114.2, 151.8, 191.1, and 260.3 µM) [189].

##### Diterpenoids

The *n*-hexane extract of *Salvia leriifolia* contains the compound 4*S*,5*R*,9*S*,10*R*-labdatrien-6,19-olide that has been reported to have an antiproliferative effect against DU145 cells (IC_50_ = 50 µM) [249]. Moreover, andrographolide, a diterpenoid from *Andrographis paniculata* has inhibited tumor growth in a mouse xenograft model with castration-resistant DU145 cells, reduced cell viability and led to apoptosis of PC-3 and DU145 cell lines [251]. The Chinese liverwort *Jungermannia fauriana* contains the ent-kaurane diterpenoids, jungermannenone A and B (JA, JB). These drugs have shown antiproliferation activities in PC-3 cells (IC_50_ = 1.34 and 4.93 μM, respectively), leading to cell apoptosis, DNA damage, mitochondrial damage, downregulated DNA repair proteins Ku70/Ku80 and RDA5, and ROS accumulation in PC-3 cells. Specifically, JA provoked marked cell cycle arrest at the G0/G1 phase (linked to c-Myc suppression), and JB forced the cell cycle block in the G2/M phase (linked to the activation of JNK signaling) [255].

##### Triterpenoids

Celastrol, an active component of *Tripterygium wilfordii* roots, notably reduced migration and proliferation of PC-3 cells, tissues invasion (VEGF secretion) in a dose-dependent manner. The pre-treatment of celastrol in a mouse model (8 μM, intratibial injection) was shown to prevent prostate cancer bone metastasis [252]. Furthermore, citral from *C. citratus* presents anticancer activity in LNCaP and PC-3 cell lines and demonstrating to be an interesting molecule for the treatment of prostate cancer (IC_50_ = 4.3 µg/mL and 14.3 µg/mL) [135]. Euphol (a tetracyclic triterpene alcohol), the main constituent of *Euphorbia tirucalli*, a subtropical and tropical plant, displays cytotoxic effects against prostate cancer cell lines (IC_50_ = 1.41–38.89 µM) [254]. Withaferin A (WA), a natural compound derived from the medicinal plant *Withania somnifera*, inhibited PC-3 cell lines growth at 1.2 µM. WA induced apoptosis via activation of caspase-8, -9 and -3, cleavage of PARP, obstructing the translocation of NF-κB and down-regulation of anti-apoptotic proteins (inhibitor of apoptosis protein - cIAP1/2, and the X-linked inhibitor of apoptosis protein) [260]. The main chemical constituent of *Fraxinus xanthoxyloides*, nummularic acid, expresses a time and dose-dependent reduction of the proliferation and colony formation capabilities of DU145 and C4-2 cells at concentrations of 5–60 μM. Nummularic acid diminished the migratory and invasive properties, amplified apoptosis, and activated 5′AMP-activated kinase (AMPK) [257]. Oenotheralanosterol B and a mixture of oenotheralanosterol A and oenotheralanosterol B from *Oenothera biennis*, exhibited antiproliferative activity against prostate cancer cell lines (IC_50_ = 8.35–49.69 μg/mL) [258]. At the concentration of 10 µg/mL, sutherlandioside D isolated from *S. frutescens* showed to restrain the Gli-reporter activity (Gli/Hh signaling) by 89% in prostate cancer cells [166]. Plectranthoic acid from *Ficus microcarpa* indicated AMPK activating properties far superior to those of metformin. Treatment of prostate cancer cells with this compound repressed proliferation and induced G0/G1 phase cell cycle arrest. Its activity was related with up-regulation of cyclin kinase inhibitors p21/CIP1 and p27/KIP1, suppressing mTOR/S6K signaling and inducing apoptosis, and autophagy (this later effect was found independent of AMPK activation) [259]. (20*R*)-Dammarane-3β,12β,20,25-tetrol (25-OH-PPD), is a ginsenoside or triterpenoid saponin isolated from *Panax ginseng*. Is a study, this drug showed anticancer activities in LNCaP and PC-3 cells (concentrations of 50 and 100 mM), by apoptosis and cell cycle arrest at G1 phase. 25-OH-PPD potently inhibit in vivo tumor growth. Moreover, a co-administration of 25-OH-PPD with taxotere, led to a 28% in tumor growth inhibition (when compared with taxotere alone), causing almost a whole inhibition of the tumor growth [250].

#### 3.2.4. Steroids

Diosgenin, a steroid compound from *Dioscorea nipponica*, shows the ability to inhibit the proliferation of DU145 cells with (IC_50_ of 6.76 µg/mL at 48 h) by activating apoptosis and autophagy (correlated with the inhibition of the PI3K/Akt/mTOR signaling pathway) [253]. Also, muricins M and N from of the fruit powder of Graviola (*Annona muricate*) have been validated for its strong anti-proliferative activities against PC-3 cells (20 µg/mL by 60%; full PC-3 cells for 24h; 80% full PC-3 cells for 47 h for muricin M) [256].

#### 3.2.5. Proteins

Diffusa cyclotides 1 to 3 (DC1-3), are peptides from *Hedyotis diffusa* (leaves and root). These compounds have showed potent cytotoxicity against LNCaP (IC_50_ 0.21–5.03 μM), PC-3 (IC_50_ 0.76–2.24 μM), DU145 (IC_50_ 0.55–3.32 μM). Particularly, DC3 inhibited the cell migration and invasion of LNCap cells, and the expansion of the nude mice tumor in prostate capan2 xenografts (1 mg/kg, inoculated subcutaneously) [246]. *Bauhinia purprea* agglutinin (BPA) is a recognized lectin that distinguishes galactosyl glycoproteins and glycolipids. BPA-PEG-modified liposomes (BPA-PEG-LP) encapsulating anticancer drugs for the treatment of prostate cancer. In a in vivo study, BPA-PEG-LP accumulated after the *i.v.* injection to DU145 solid cancer-bearing mice, and intensely bound to the cancer cells. Also, the same mice *i.v.* injected with BPA-PEG-LP encapsulating doxorubicin (BPA-PEG-LPDOX,) or PEG-modified liposomes encapsulating DOX (PEG-LPDOX), BPA-PEG-LPDOX considerably inhibited the growth of the DU145 cancer cells, while PEG-LPDOX exhibited little anticancer effect [245]. DLasiL, a lectin isolated from seeds of the *Dioclea lasiocarpa* indicated highly potent antiproliferative activity against PC-3 (low nanomolar) and was as high or more potent than the lectins ConBr (*Canavalia brasiliensis*), ConM (*Canavalia maritima*) and DSclerL (*Dioclea sclerocarpa*) against the same cells [247].

#### 3.2.6. Fatty Acids

(*E*)-ethyl 8-methylnon-6-enoate from *C. chinense* showed antiproliferative effects on LNCaP, DU145 and PC-3 lines (76%, 41% and 62% at 100 μM, respectively) [123]. Juglone from plants inhibits the migration and invasion of LNCaP and LNCaP-AI cell lines, increasing the expression of the epithelial marker E-cadherin while declining the expression of mesenchymal markers (N-cadherin and vimentin) in a dose-dependent manner. This compound also suppresses EMT via the Akt/GSK-3β/Snail pathway, therefore diminishing the invasiveness of prostate cancer cells [223].

## 4. Evidence from Clinical Studies

Prostate cancer patients are progressively using complementary and alternative medicines in order to support the immune system in addition to conventional treatments (Table 5). This minimizes morbidity related to conventional treatments, enhances the quality of life, eventually, in the hope of finding a cure when conventional treatment fails [265].

In a retrospective study, danshen (*Salvia miltiorrhiza*) showed in vivo protective effects in prostate cancer patients. A total of 40,692 prostate cancer patients were followed for 15 years. The survival rate analyses confirmed a strong dose-dependent and time-dependent connection between the use of danshen and survival (an increase of 5–10%), supporting the protective effects of danshen on prostate cancer patients. These results might serve as a base for a new therapeutic approach for the treatment of prostate cancer [266]. Another retrospective cohort study, between 1998 and 2003 in Taiwan, organized patients as traditional Chinese medicine (TCM) users or nonusers, revealed that Chai-Hu-Jia-Long-Gu-Mu-Li-Tang was the most significant TCM formulae for improving survival in metastatic prostate cancer patients (adjusted hazard ratio 0.18). The formula was later approved by the Ministry of Health and Welfare, Taiwan [267].

Pantuck, et al. [268] described the first clinical trial of pomegranate juice in prostate cancer patients. In a placebo-controlled study, pomegranate juice consumption showed to have a noteworthy extension of PSA doubling time (PSADT), coupled with consistent prostate cancer in vitro cell effects on proliferation, apoptosis, and oxidative stress [268]. In phase II clinical trial, a prolongation of PSA doubling following consumption was exposed, and there were indicators of oxidative stress enhanced in human subjects [269]. These results were corroborated in another phase II study, in men receiving pomegranate seed extract [270]. In fact, two doses of pomegranate extract (POMx) treatment was correlated with a ≥ 6 month rise in PSADT in both treatments, with no adverse effects detected. However, it is important to note that the implication of this in-study reduction of PSADT remains uncertain and a placebo-controlled study in this patient population is still required [270]. In another study, compared with placebo, pomegranate extract did not significantly extend PSADT in prostate cancer patients with high PSA, after primary therapy. It was noted that men with the manganese superoxide dismutase (MnSOD) AA genotype might embody a more sensitive group to the antiproliferative effects of pomegranate on PSADT; however, this finding needs more testing and validation [160].

Polyphenol-rich foods such as pomegranate, green tea, broccoli, and turmeric have proven to possess anti-neoplastic activities in laboratory models. In a randomized controlled trial, 199 men, average age 74 years-old, with localized prostate cancer, were randomized to receive an oral capsule containing a blend of pomegranate, green tea, broccoli, and turmeric, or an identical placebo for 6 months. Results revealed that the median rise in PSA in the food supplement group was 14.7%, in contrast to 78.5% in the placebo group. No significant differences within the predetermined subgroups (e.g., age, treatment category, body mass index, cholesterol, blood pressure, blood sugar, C-reactive protein or adverse events), however more evaluations are required to validate these results [271].

Diets high in cruciferous vegetables are correlated with a lower risk of incidence and aggressiveness of prostate cancer. A clinical trial in 20 men with recurrent prostate cancer, receiving 200 μM/day of sulforaphane-rich extracts for a maximum period of 20 weeks, exhibited a general increase of the on-treatment PSADT (6.1 months) compared with the pre-treatment PSADT (9.6 months). No grade 3 adverse events were described, but a further trial at higher doses is needed to verify the role of these extracts in the prevention or treatment of prostate cancer [272].

In a randomized placebo-controlled clinical study, two doses of resveratrol (150 mg or 1000 mg resveratrol daily) was administered for 4 months. The results showed that resveratrol decreases the levels of circulating androgen precursors, nonetheless with no effect on testosterone, dihydro-testosterone, PSA levels or prostate volume in middle-aged men [273]. In a phase I clinical trial, 14 men with recurrent prostate cancer received different doses of resveratrol of MSKE, a pulverized muscadine grape (*V. rotundifolia*) consisting in 4.4 μg resveratrol/500 mg MSKE for a period of 2–31 months. The highest MSKE dose (4000 mg) was confirmed safe and able to elongate the PSADT with a median of 5.3 months. Both low-dose (500 mg) and high-dose (4000 mg) MSKE are being further explored in a randomized, multicenter, placebo-controlled, dose-evaluating phase II trial [274].

PC-SPEC, a comprehensive study of a Chinese formula, consisting of eight medicinal plants, was applied in 69 patients with prostate cancer; they were treated with 320 mg/3 times a day. The outcome revealed that 82% decreased PSA in 2 months, 78% in 6 months and 88% in 12 months post-treatment in some side effects (e.g., phlebitis) [275].

## 5. Conclusions and Future Perspectives

The present report suggests that several medicinal plants, herbs, and isolated phytochemicals possess interesting potential both in the prevention and even treatment of prostate cancer. Although multiple plant species have shown interesting biological attributes, it is worth mentioning that it is their chemical constituents that have been shown to play a preponderant role in their final bioactivity. Phenolic compounds are ones that have revealed the more promising potential, followed by alkaloids and terpenoids. Anyway, and despite encouraging data seen in some of these interventions, only randomized controlled trials can deliver adequate evidence to create universal guidelines. In addition, unfortunately, only a scarce number of phytochemicals have been undergone clinical trials to assess their anti-prostate cancer potential. The lack of clinical evidence to support the in vitro and in vivo results need further clinical studies of the widespread use of the reported ingredients for chemoprevention and therapy of prostate cancer.

## Figures and Tables

**Figure 1 nutrients-11-01483-f001:**
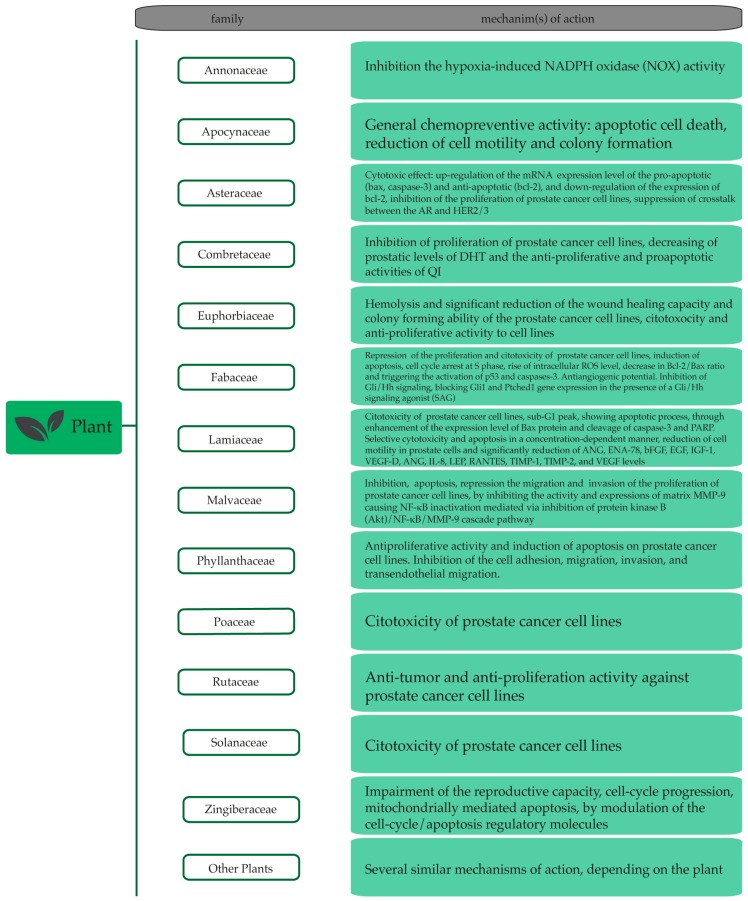
Plant species with anti-prostate cancer potential and its respective modes of action.

**Table 1 nutrients-11-01483-t001:** Relative risk of prostate cancer in patients with a positive family history.

Risk Group	Relative Risk of Prostate Cancer
Father and brother had prostate cancer	9
≥2 first degree relatives having prostate cancer	4.39
Brothers having prostate cancer	3.14
First degree relative with prostate cancer at the age of<65	2.87
Second degree relative with prostate cancer	2.52
One first degree relative with prostate cancer	2.48
Father having prostate cancer	2.35
First degree relative with prostate cancer at the age of ≥65	1.92

**Table 2 nutrients-11-01483-t002:** Classification of the risk groups of prostate cancer [83].

Risk Group	Clinical Stage	PSA (ng/mL)	Gleason Score	Biopsy Criteria
Low	T1a or T1c	<10	2–6	Unilateral or <50% of core involved
Intermediate	T1b, T1c, or T2a	<10	3 + 4 = 7	Bilateral
High	T1b, T1c, T2b, or T3	10–20	4 + 3 = 7	>50% of core involved or perineural invasion or ductal differentiation
Very high	T4	>20	8–10	Lymphovascular invasion or neuroendocrine differentiation

**Table 3 nutrients-11-01483-t003:** Medicinal plants with anti-prostate cancer effects.

Plant Species	Family	In Vitro	In Vivo	References
*Acacia catechu*	Fabaceae	+	-	[110]
*Achillea santolinoides*	Asteraceae	+	*-*	[111]
*Achillea teretifolia*	Asteraceae	+	*-*	[112]
*Allium wallichii*	Amaryllidaceae	+	-	[113]
*Aloe perryi*	Xanthorrhoeaceae	+	*-*	[114]
*Anaxagorea brevipes*	Annonaceae	+	-	[115]
*Angelica gigas*	Apiaceae	-	+	[116,117]
*Annona muricata*	Annonaceae	+	*-*	[118]
*Anogeissus latifolia*	Combretaceae	+	-	[110]
*Apocynum venetum*	Apocynaceae	+	-	[119]
*Arachis hypogaea*	Fabaceae	+	*-*	[120]
*Baliospermum montanum*	Euphorbiaceae	+	+	[121]
*Berberis libanotica*	Berberidaceae	+	-	[122]
*Byrsonima crassifolia*	Malpighiaceae	+	*-*	[123]
*Calliandra portoricensis*	Fabaceae	+	-	[124]
*Capsicum chinense*	Solanaceae	+	*-*	[123]
*Carica papaya*	Caricaceae	+	-	[125]
*Cascabela peruviana*	Apocynaceae	+	*-*	[126]
*Chenopodium hybridum*	Amaranthaceae	+	*-*	[127]
*Cnidoscolus chayamansa*	Euphorbiaceae	+	*-*	[123]
*Cornus mas*	Cornaceae	+	-	[128]
*Costus pulverulentus*	Costaceae	+	*-*	[129]
*Crataegus Pinnatifida*	Rosaceae	+	-	[130]
*Crocus sativus*	Iridaceae	+	+	[131,132,133]
*Curcuma longa*	Zingiberaceae	+	*-*	[131,134]
*Cymbopogon citratus*	Poaceae	+	*-*	[135]
*Cymbopogon giganteus*	Poaceae	+	*-*	[135]
*Euphorbia microsciadia*	Euphorbiaceae	+	*-*	[111]
*Euphorbia szovitsii*	Euphorbiaceae	+	*-*	[111]
*Eurycoma longifolia*	Simaroubaceae	+	+	[136]
*Fagara zanthoxyloides*	Rutaceae	+	-	[137]
*Fagopyrum esculentum*	Polygonaceae	+	*-*	[138]
*Fagopyrum tataricum*	Polygonaceae	+	*-*	[138]
*Ficus deltoidea* var. *angustifolia*	Moraceae	+	*-*	[139]
*Ficus deltoidea* var. *deltoidea*	Moraceae	+	*-*	[139]
*Formosa lambsquarters*	Amaranthaceae	+	*-*	[138]
*Glycine max*	Fabaceae	+	-	[140]
*Glycyrrhiza uralensis*	Fabaceae	+	*-*	[141]
*Haplophyllum perforatum*	Rutaceae	+	*-*	[111]
*Helicteres hirsuta*	Malvaceae	+	*-*	[142]
*Hertia angustifolia*	Asteraceae	+	*-*	[111]
*Hibiscus sabdariffa*	Malvaceae	+	+	[143]
*Leucaena leucocephala*	Fabaceae	+	*-*	[123]
*Lysimachia ciliata*	Primulaceae	+	-	[144]
*Malmea depressa*	Annonaceae	+	*-*	[123]
*Maytenus royleana*	Celastraceae	+	+	[145]
*Medicago sativa*	Fabaceae	+	*-*	[111]
*Melissa officinalis*	Lamiaceae	+	-	[146,147]
*Mentha arvensis*	Lamiaceae	+	-	[148]
*Mentha spicata*	Lamiaceae	+	-	[148]
*Mentha viridis*	Lamiaceae	+	-	[148]
*Moringa oleifera*	Moringaceae	+	-	[110]
*Nepeta cataria*	Lamiaceae	+	*-*	[149]
*Nigella sativa*	Ranunculaceae	+	*-*	[131,150]
*Oryza sativa*	Poaceae	+	-	[151]
*Paeonia lactiflora*	Paeoniaceae	+	-	[152].
*Paramignya trimera*	Rutaceae	+	-	[153]
*Phyllanthus amarus*	Phyllanthaceae	+	-	[154]
*Phyllanthus niruri*	Phyllanthaceae	+	-	[154]
*Phyllanthus urinaria*	Phyllanthaceae	+	-	[154]
*Phyllanthus watsonii*	Phyllanthaceae	+	-	[154]
*Plumbago zeylanica*	Plumbaginaceae	+	-	[155]
*Polygonatum* sp	Asparagaceae	+	*-*	[156]
*Pseudocedrela kotchyi*	Meliaceae	+	-	[137]
*Psidium guajava*	Myrtaceae	+	*+*	[138,157,158]
*Punica granatum*	Lythraceae	+	+	[5,159,160,161]
*Quisqualis indica*	Combretaceae	+	+	[162]
*Remotiflori radix*	Campanulaceae	+	+	[163]
*Salvia multicaulis* Vahl	Lamiaceae	+	*-*	[111]
*Salvia trilobal*	Lamiaceae	+	-	[164]
*Sigesbeckia orientalis*	Asteraceae	+	-	[165]
*Sophora alopecuroides*	Fabaceae	+	*-*	[111]
*Sutherlandia frutescens*	Fabaceae	+	+	[166]
*Terminalia bellerica*	Combretaceae	+	-	[110]
*Terminalia catappa*	Combretaceae	+	*-*	[123]
*Urtica dioica*	Urticaceae	+	*-*	[111,167]
*Vitis rotundifolia*	Vitaceae	+	-	[168]
*Wedelia chinensis*	Asteraceae	-	+	[169,170]
*Withania coagulans*	Solanaceae	*-*	+	[171]
*Xylopia aethiopica*	Annonaceae	+	-	[172]
*Zanthoxyli fructus*	Rutaceae	+	+	[173]
*Zingiber officinale*	Zingiberaceae	+	+	[131,174,175]

+: Showed in vitro or in vivo antiproliferative effect; -: Not found.

**Table 4 nutrients-11-01483-t004:** Plant derived-compounds with anti-prostate cancer effects.

Bioactive Compounds	In Vitro	In Vivo	References
**Alkaloids**			
(−)-Anonaine	+	-	[182]
(−)-Caaverine	+	-	[182]
(−)-Nuciferine	+	-	[182]
6-Hydroxycrinamine	+	-	[183]
7-Hydroxydehydronuciferine	+	-	[182]
Capsaicin	+	-	[184]
Crinamine	+	-	[183]
Emetine	+	+	[185,186]
Liriodenine	+	-	[182]
Lycorine	+	+	[183,187]
Matrine	+	-	[188]
Oxymatrine	+	-	[188]
Oxysophocarpine	+	-	[188]
Schisanspheninal A	+	-	[189]
Sophocarpine	+	-	[188]
Tetrandrine	+	-	[190]
**Carotenoids**			
Crocetin	+	-	[133]
Crocin	+	-	[132]
**Fatty acid**			
(*E*)-ethyl 8-methylnon-6-enoate	+	-	[123]
**Phenolic compounds**			
α-Mangostin	+	+	[191].
γ-Tocopherol	+	-	[192]
δ-Tocotrienol	+	-	[192]
(-)-5,7-Difluoroepicatechin-3-*O*-gallate	+	-	[193]
(-)-Epicatechin-3-*O*-gallate	+	-	[193]
10-Gingerol	+	-	[175]
6-Gingerol	+	-	[175]
6-Prenylnaringenin	+	-	[194]
6-Shogoal	+	-	[175]
7-*o*-Galloyl catechin	+	-	[195]
8-Gingerol	+	-	[175]
8-Prenylnaringenin	+	-	[194]
Afzelin	+	-	[196]
Altholactone	+	-	[197]
Apigenin		+	[198]
Camptothin B	+	-	[141]
Catechin	+	-	[195]
Catechin-3-*o*-gallate	+	-	[195]
Chlorogenic acid	+	-	[130]
Chrysin	+	-	[199]
Cinnamaldehyde	+	-	[200]
Cornusiin A	+	-	[141]
Cornusiin H	+	-	[141]
Curcumin	+	+	[201,202,203,204]
Decursin	+	-	[117]
Decursinol angelate	+	-	[117]
Dehydrozingerone	+	-	[205]
Delphinidin	+	+	[206,207]
Ellagic acid	+	+	[208,209]
Eugenol	+	-	[200]
Fisetin	+	+	[210]
Flavokawain A	+	+	[211]
Flavopiridol	+	+	[212]
Garcinol	+	+	[213,214]
Ginkgetin	+	+	[215]
Hesperetin	+	-	[216]
Hirsutenone	+	-	[217]
HLBT-100 or HLBT-001 (5,3′-dihydroxy- 6,7,8,4′-tetramethoxyflavanone)	+	-	[218]
Honokiol	+	-	[219]
Icarisid II	+	-	[220]
Isoangustone A	+	-	[221,222]
Isovitexin	+	-	[139]
Juglone	+	-	[223]
Licoricidin	+	-	[221,222]
Magnolol	+	-	[224]
Mangiferin	+	+	[225,226]
Maysin	+	-	[227]
Methyl gallate	+	-	[195]
Osthol	+	-	[4,228]
Oxyfadichalcones A	+	-	[229]
Oxyfadichalcones B	+	-	[229]
Oxyfadichalcones C	+	-	[229]
Oxyfadichalcones D	+	-	[229]
Oxyfadichalcones E	+	-	[229]
Oxyfadichalcones F	+	-	[229]
Oxyfadichalcones G	+	-	[229]
Paeonol	+	+	[230]
Peperotetraphin	+	-	[231]
Physangulatins I	+	-	[232]
Plumbagin	+	+	[155,233]
Punicalagin	+	-	[234]
Quercetin	+	+	[235,236,237]
Resveratrol	+	+	[238,239,240]
Rutin	+	-	[241]
Tannic acid	+	-	[242]
Tricin	+	-	[243]
Xanthohumol	+	-	[188,244]
**Protein**			
Agglutinin	+	+	[245]
Diffusa cyclotide 1	+	-	[246]
Diffusa cyclotide 2	+	-	[246]
Diffusa cyclotide 3	+	+	[246]
Lectin ConBr	+	-	[247]
Lectin ConM	+	-	[247]
Lectin DLasiL	+	-	[247]
Lectin DSclerL	+	-	[247]
**Terpenoids**			
α-Santalol	+	+	[248]
4*S*,5*R*,9*S*,10*R*-Labdatrien-6,19-olide	+	-	[249]
(20*R*)-Dammarane-3β,12β,20,25-tetrol (25-OH-PPD)	+	+	[250]
Andrographolide	+	+	[251]
Celastrol	+	+	[252]
Citral	+	-	[135]
Diosgenin	+	-	[253].
Euphol	+	-	[254]
Isocuparenal	+	-	[189]
Jungermannenone A	+	-	[255]
Jungermannenone B	+	-	[255]
Muricins M	+	-	[256]
Muricins N	+	-	[256]
Nummularic acid	+	-	[257]
Oenotheralanosterol B	+	-	[258]
Plectranthoic acid	+	-	[259]
Sutherlandioside D	+	-	[166].
Widdaranal A		-	[189]
Widdaranal B	+	-	[189]
Widdarol peroxide	+	-	[189]
Withaferin A	+	-	[260]

-, no effect observed; +, positive effect.

**Table 5 nutrients-11-01483-t005:** Clinical trials showing the anti-prostate cancer potential of plant-derived phytochemicals.

Phytochemicals/Formulae	Bioactive Effect	Reference
Danshen (*Salvia miltiorrhiza*)	Protective effects; Improved survival (5–10%)	[266]
TCM formulae (Chai-Hu-Jia-Long-Gu-Mu-Li-Tang)	Improved survival	[267]
Pomegranate juice	Extension of PSA doubling time, with no adverse effects	[268,269,270]
Pomegranate, green tea, broccoli, turmeric	Decreased PSA levels	[271]
Resveratrol	Decreased the circulating levels of androgen precursors	[273]
	Extension of PSA doubling time, with no adverse effects	[274]
PC-SPEC	Decreased PSA levels	[275]

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
