# Peer review of "Phytochemicals in Prostate Cancer: From Bioactive Molecules to Upcoming Therapeutic Agents"

_nutrients, 2019, doi:10.3390/nu11071483_

Round 1
Reviewer 1 Report
The review article is interesting.
The authors could describe the conclusions and future perspectives of the phytochemicals in greater detail.
A figure illustrating the various uses of the phytochemicals and their targets would be more illustrative.
Author Response
The review article is interesting.
Answer: Thank you for the overall appreciation of our work
The authors could describe the conclusions and future perspectives of the phytochemicals in greater detail.
Answer: Conclusions and future perspectives section was improved
A figure illustrating the various uses of the phytochemicals and their targets would be more illustrative.
Answer: A proper figure was included
Reviewer 2 Report
The review article is well written however there are several recommendations-
1. Please include a table for all the phytochemicals that are in clinical trial.
2. Few important phytochemicals such as 6-Shogaol, curcumin and resveratrol that showed promising in vivo effect in mouse model should be described. The author might consider PMID:24691500 and other papers of 6 Shogaol.
3. It would have been nice to have a table with combination of phytochemicals that showed efficacy in Prostate cancer. Example include (PMID: 29202102).
Author Response
The review article is well written however there are several recommendations-
Answer: Thank you for the overall appreciation of our work
Please include a table for all the phytochemicals that are in clinical trial.
Answer: A proper table was included
2. Few important phytochemicals such as 6-Shogaol, curcumin and resveratrol that showed promising in vivo effect in mouse model should be described. The author might consider PMID:24691500 and other papers of 6 Shogaol.
Answer: This paper, although described plant-derived phytochemicals with anti-prostate cancer potential, its main focus was to assess and to deliver data from plant species towards to incite more in-depth studies.
3. It would have been nice to have a table with combination of phytochemicals that showed efficacy in Prostate cancer. Example include (PMID: 29202102).
Answer: As referred previously, we summarized in a table all the phytochemicals to which several studies were already performed to assess its anti-prostate cancer potential. However, the clinical efficacy from pre-clinical to clinical studies using plant-derived phytochemicals is the focus of another review article, since exploring this aspect in this work would becomes this review extremely heavy.
Round 2
Reviewer 2 Report
No further comments